

# The Community Cloud retrieval for Climate (CC4CL). Part I: A framework applied to multiple satellite imaging sensors

Oliver Sus[1], Martin Stengel[1], Stefan Stapelberg[1], Gregory McGarragh[2], Caroline Poulsen[3], Adam C. Povey[4], Cornelia Schlundt[1], Gareth Thomas[3], Matthew Christensen[2,3], Simon Proud[2], Matthias Jerg[1], Roy Grainger[4], and Rainer Hollmann[1]

[1]DWD - Deutscher Wetterdienst, Frankfurter Straße 135, 63067 Offenbach, Germany
[2]Department of Physics, University of Oxford, Clarendon Laboratory, Parks Road, Oxford OX1 3PU, U.K.
[3]RAL Space - Rutherford Appleton Laboratory, Chilton, Didcot, OX11 0QX, U.K.
[4]National Centre for Earth Observation, Atmospheric, Oceanic and Planetary Physics, University of Oxford, Parks Road, Oxford OX1 3PU, U.K.

*Correspondence to:* Martin Stengel
martin.stengel@dwd.de

**Abstract.** We present the key features of the Community Cloud retrieval for CLimate (CC4CL) processing algorithm. We focus on the novel features of the framework: the optimal estimation approach in general, explicit uncertainty quantification through rigorous propagation of all known error sources into the final product, and the consistency of our long-term, multi-platform time-series provided at various resolutions, from 0.5° to 0.02°.

By describing all key input data and processing steps, we aim to inform the user about important features of this new retrieval framework, and its potential applicability to climate studies. We provide an overview of the retrieved and derived output variables. These are analysed for four, partly very challenging, scenes collocated with CALIOP (Cloud-Aerosol lidar with Orthogonal Polarization) observations in the high-latitudes and over the Gulf of Guinea/West Africa.

The results show that CC4CL provides very realistic estimates of cloud top height and cover for optically thick clouds but,

where optically thin clouds overlap, returns a height between the two layers. CC4CL is a unique, coherent, multi-instrument cloud property retrieval framework applicable to passive sensor data of several EO missions. Through its flexibility, CC4CL offers the opportunity for combining a variety of historic and current EO missions into one data set, which, compared to single sensor retrievals, is improved in terms of accuracy and temporal sampling.

## 1   Introduction

Satellite data are an essential source of information for understanding and predicting climate change. They provide global long-term observations from which geophysical parameters can be derived. These are used for time-series analysis of climate variables, and also for the assimilation into or validation of climate models (Comiso and Hall, 2014; Yang et al., 2013). A paramount goal of these efforts is the comprehensive characterization of the global energy and water budgets (Stephens et al., 2012).



Clouds considerably influence the global energy budget through shielding and direct forcing effects (Kiehl and Trenberth, 1997). However, clouds are difficult to quantify, having highly variable composition and spatiotemporal distributions, and produce the largest uncertainty in our understanding of climate change (Norris et al., 2016; IPCC, 2013). Observations from passive imagers do not sufficiently resolve several important cloud properties, such as vertical structure, sub-pixel hetero-
geneity, the cloud boundary, and the column-integrated ice or liquid water path. Several secondary variables (state of surface and atmosphere, viewing geometry, sensor calibration and spectral response uncertainties) further complicate cloud retrievals, propagating uncertainties into the derived cloud properties (Hamann et al., 2014). Nonetheless, passive satellite imagers are the most widely used instruments for cloud retrievals as they provide long-term, global coverage at acceptable cost for the user.

There are several satellite-based retrieval frameworks. One of the earliest is the International Cloud Climatology Project
(ISCCP) (Rossow and Schiffer, 1999). ISCCP provides data on cloud products for 1983–2009, and introduced a cloud type classification based on cloud optical thickness-cloud top pressure (COT-CTP) joint histograms that is still popular even today. Continuously reprocessed retrieval systems include Pathfinder Atmosphere Extended (PATMOS-x) (Heidinger and Pavolonis, 2009; Heidinger et al., 2012), EUMETSAT Satellite Application Facility on Climate Monitoring (CM SAF) cLoud, Albedo and RAdiation (CLARA-A1) (Karlsson et al., 2013), and MODIS Collection 6 (MODIS C6) (Platnick et al., 2017). These
retrievals vary in their auxiliary data sources, approaches, and complexity but generally use radiative transfer models and/or derived look-up tables (LUT) to provide a clear-sky reference and for simulating atmospheric and cloud contributions to top of atmosphere (TOA) radiances. Cloud properties are derived using decision trees and thresholding (PPS in CLARA-A1), LUT based inversions (MODIS C6), or optimal estimation theory (PATMOS-X). COT and CER (cloud effective radius) are usually calculated following Nakajima and King (1990). However, the derived microphysical variables are not guaranteed to be
radiatively consistent with independently derived cloud parameters. For cloud masking, the retrieval frameworks apply various approaches such as Naïve Bayes (PATMOS-X), dynamic thresholding (CLARA-A1), or a battery of threshold tests (MODIS C6). Finally, cloud phase or type is determined as a function of a combined convergence/cloud top temperature (CTT)-test (CLARA-A1), the Pavolonis et al. (2005) threshold algorithm (PATMOS-X), or a bispectral decision tree considering channels at 8.5 and 11 μm (MODIS C6). Compared to AVHRR, MODIS has several additional spectral channels that provide cloud
microphysical information (Platnick et al., 2017), such that MODIS data provide more information for retrieving cloud products than AVHRR. Still, the MODIS C6 cloud top retrieval loses sensitivity for optically thinner clouds (COT < 2, Menzel et al. (2010); Christensen et al. (2013)), and sees into the cloud to an optical thickness of approximately unity (Baum et al., 2012). This complicates validation against independent measurements such as those derived from lidar, which explicitly observe the cloud top. Despite some promising results, these studies show that current retrievals underestimate cloud top pressure for
optically thin clouds even when the full potential of MODIS spectral coverage is used.

There are numerous studies that evaluate the performance of the aforementioned retrievals for cloud cover with weather station data, such as over the Mediterranean (Sanchez-Lorenzo et al., 2017) and conterminous United States (Sun et al., 2015). The results are variable, but generally show that the inter-annual correlation is highest for PATMOS-X (up to r = 0.94) and lowest for CLARA-A1 (r = 0.20 – 0.7). More importantly, these studies emphasize the difficulty of deriving reliable cloud cover
trends from AVHRR time series, as the retrievals overestimate the change in cloud cover by as much as an order of magnitude



(Sun et al., 2015). There are also several evaluation or validation studies for individual retrieval algorithms. Differences between PATMOS-X microphysical retrievals using MODIS data and the collocated MYD06 product are within retrieval uncertainty (Walther and Heidinger, 2012). CLARA-A2 underestimates global cloud top height (CTH) by 840 m compared to CALIOP. Comparing CLARA-A2 to PATMOS-X, MODIS C6 and ISCCP, it underestimates global CTP by 4–90 hPa and has a cloud
phase bias of lower than 9 % (Karlsson et al., 2016). MODIS C6 CTH bias for low-level boundary layer water clouds is 197 m compared to CALIOP, and the phase detection has been improved for optically thin ice clouds. However, the detection of supercooled water clouds remains problematic (Baum et al., 2012). For MODIS C5, global CTH was underestimated relative to CALIOP by 1.4 km (Holz et al., 2008).

     Satellite observations of clouds are available for the last 40 years. However, data need to be carefully processed and analysed
in order to derive a consistent long-term data record from several inter-calibrated satellite platforms. Consistency can be traded for continuity, and multi-platform algorithms could exploit additional data when newer sensors become available. Modern sensors provide improved spectral coverage and spatial resolutions and, thus, potentially better cloud retrievals. However, their data records are too short to produce climatologies of at least 30 years, and discontinuities are built into time series when higher resolution satellite data are input to the processing. Major complications of cloud retrievals include optically
transparent clouds, multi-layer or overlapping clouds, and effective cloud top height determination. The degree to which these complications can be addressed depends on the nature of the retrieval and the type of input satellite data used. MODIS provides a much larger spectral sampling than the six AVHRR heritage channels. MODIS and atmospheric sounders are clearly superior when detecting cloud height through the application of the "CO2-slicing" technique. However, when consistent climatologies are to be built, time series length and spatiotemporal resolution limit the choice in retrieval type and input satellite data.

The European Space Agency has established the ESA Climate Change Initiative program (ESA CCI, 2015; Hollmann et al., 2013) in order to advance knowledge of the climate system. The project's primary focus is the production of thirteen Essential Climate Variables (ECVs) for ocean, atmosphere, and land. The main objective of ESA Cloud_cci is to develop a state-of-the-art open-source community cloud retrieval algorithm which is capable of processing passive imager data for a number of (non-)European satellites covering several decades. We used satellite data from MODIS Aqua and Terra (2000–2014) (King
et al., 1992), AVHRR on NOAA-7 to NOAA-19 and METOPA (1978–2014) (Jacobowitz et al., 2003), ATSR-2 on ERS-2 (1995–2003), and AATSR on ENVISAT (2002–2012). Only the AVHRR-equivalent channels from MODIS and AATSR are used. Hence, the resulting retrieval data are hereafter referred to as the "AVHRR heritage dataset". Moreover, the resulting time series are carefully validated against well-established climatologies (ISCCP, PATMOS-x, CM SAF, and MODIS Collection 6), reanalysis and model data (ERA-Interim and EC-Earth), ground-truth synoptic observations, and CALIOP lidar data.

The CC4CL core algorithm was developed in a modular fashion and provides open-source access to support distribution and development within the scientific community. Particular attention was paid to allow processing of multiple instruments within a single framework, thus maximising the consistency of cloud products independent of the sensor source. The framework accounts for physical consistency amongst all output variables and radiative consistency amongst all input satellite radiances. This is an improvement on other established retrieval frameworks. These commonly derive COT and CER by adopting the
Nakajima and King (1990) approach, but macrophysical products are estimated independently and are thus radiatively in-





consistent with the former variables. Another novel feature of CC4CL is the production of uncertainty estimates of retrieval parameters through explicit error propagation from input to output data. With these criteria in mind, the Optimal Retrieval of Aerosol and Cloud (ORAC) (Thomas et al., 2009a; Poulsen et al., 2012) was chosen from three competing algorithms within a "Round Robin" selection process (Stengel et al., 2015).

In this study, we present the key features of the CC4CL processing algorithm. We particularly focus on discussing the novel features of the framework: the optimal estimation approach in general, the explicit uncertainty quantification through rigorous propagation of all known error sources to the final product, and the consistency of our long-term, multi-platform time-series provided at various resolutions, from 0.5° to 0.02°. By describing all key input data and processing steps, we inform the future user about important features of this new processing framework, and its potential applicability in climate studies. We provide
an overview of the retrieved and derived output variables. These are initially validated in a comprehensive and detailed analysis of retrieval results that we collocated with CALIOP observations for three scenes in the Arctic and one scene in the Gulf of Guinea/West Africa. The results show that CC4CL produces mixed-layer estimates for cases where optically thin clouds overlap, but provides very realistic estimates of cloud top height and cover for optically thick clouds.

## 2    Data and methods

### 2.1    L1 satellite data

#### 2.1.1    AVHRR

The Advanced Very High Resolution Radiometer (AVHRR) is a cross-track scanner with a 2900 km swath width, providing almost daily global coverage. The sensor is equipped with six spectral channels (Table 01), out of which only five can be transmitted simultaneously so that either channel 3a or 3b is available. In-flight calibration is performed only for thermal
channels, using a stable blackbody and a space view as references. AVHRR has been mounted on several NOAA platforms as well as on EUMETSAT's MetopA/B, all of which are sun-synchronous, polar orbiting satellites. Due to a lack of orbit control technology for all NOAA AVHRR's, there is considerable orbit drift in equatorial crossing times (ECT) both for morning (ECT < 12:00 LST) and afternoon (ECT > 12:00 Local Solar Time (LST)) satellites. To reduce drift-induced changes in retrieved cloud properties, any AVHRR is replaced with its corresponding successor once available (= the AVHRR prime record).
Typically, one morning and one afternoon NOAA satellite are in orbit at any time.

For CC4CL, we use Global Area Coverage (GAC) L1c data on a reduced spatial resolution of 1.1 km × 4 km at nadir (Devasthale et al., 2017). The AVHRR GAC L1c data record, including advanced inter-calibration efforts, was produced for ESA Cloud_cci and CMSAF (Schulz et al., 2009; Karlsson et al., 2013). CC4CL processed AVHRR data from 08/1981 (NOAA-7) up to 12/2014 (MetopA + NOAA-19). We applied a filtering technique to channel 3b data, and a database algorithm for splitting
midnight orbits and blacklisting.



### 2.1.2 MODIS

The Moderate Resolution Imaging Spectroradiometer (MODIS) is carried by NASA's Terra and Aqua satellite platforms in a near sun-synchronous polar orbit at 705 km altitude. Due to orbit control, ECT is a constant 10:30 LST for Terra, and 13:30 LST for Aqua. The Aqua satellite is a member of the "A-Train" constellation, which also includes the CALIOP and CloudSat

satellites. MODIS is a cross-track scanner with a 2330 km swath width, producing a complete near-global coverage in less than two days (Xiong et al., 2009).

CC4CL is applied to Collection 6 MOD021km (Terra) and MYD021km (Aqua) L1b input data (NASA LP DAAC, 2015). For the AVHRR-heritage dataset produced here, the NASA Goddard space flight centre performed a spectral subsetting of the 36 MODIS channels available (see Table 01 for the channels extracted), and data were directly shipped to ECMWF (European

Centre for Medium-Range Weather Forecasts) for archiving. The files are stored in HDF-EOS format at 1km spatial resolution, with the 250 m and 500 m channels having been aggregated to 1 km resolution. MODIS L1b data are organized in granules, each of which contains ~5 minutes of MODIS data or ~203 scan lines. Geolocation information is provided in separate files for Terra (MOD03) and Aqua (MYD03), containing geodetic latitude and longitude and solar/satellite zenith and azimuth angles. L1b data are corrected for all known instrumental effects through on-board calibrator data, and are organized into a viewing

swath matching the geolocation file structure (MODIS Characterization Support Team, 2009). With CC4CL, we processed data from 02/2000 (Terra) or 08/2002 (Aqua) to 12/2014.

### 2.1.3 ATSR-2 and AATSR

The second and third generation Along Track Scanning Radiometers (ATSR-2 and Advanced ATSR, Merchant et al. (2012)) were launched on ESA's polar orbiting satellites ERS-2 and ENVISAT in 04/1995 and 03/2002, respectively. Both platforms

were put into a sun-synchronous orbit at ~780 km altitude, with ECT = 10:30 LST for ERS-2 and ECT = 10:00 for ENVISAT. Both ATSRs are identical in their overall configuration except for data transfer bandwidth (Table 01). ATSR is designed to be self-calibrating, with two on-board black-body targets for the thermal channels and a sun-illuminated opal target for the visible/near-infrared channels. ATSR uses a dual-view system: a nadir view, and a forward view scanning the surface at an angle of 55°. The continuous scanning pattern produces a nadir resolution of approximately 1 km × 1 km with a swath width

of 512 pixels or ~500 km, providing global coverage every six days.

We used no forward view data for cloud retrievals, as the 3-dimensional cloud structure produces parallax effects which are not accounted for within the current forward model. With CC4CL, we processed ATSR data from launch date until 05/2003 (ERS-2) and 04/2012 (ENVISAT).



## 2.2 Auxiliary data

### 2.2.1 ERA-Interim

We use ERA-Interim data as first-guess input for the retrieval of surface temperature, and as input for the neural network cloud mask. ERA-Interim is a reanalysis of the global atmosphere, and is available from 1979 until today (Berrisford et al., 2011; Dee et al., 2011). The atmospheric profile variables are defined at 60 vertical levels. The original horizontal resolution is defined through a T255 spherical-harmonic representation for the basic dynamical fields, and through a reduced Gaussian grid with ~79 km spacing for surface fields. We downloaded ERA-Interim data from the ECMWF's MARS archive at a spatial resolution of 0.72°(the default preprocessing grid resolution), and at a higher resolution of 0.1° for the neural network cloud mask input variables (Table A1). We acquired analysis (i.e. not forecast) data at 6-hourly timesteps. After download, all files were remapped to the CC4CL preprocessor grid through Climate Data Operators (CDO, 2015). This was necessary, as ERA-Interim coordinates are defined at the cell boundaries, whereas they are defined at the cell centres within CC4CL. The reanalysis data are temporally interpolated onto the satellite image's centre time by linearly weighting the files before and after.

ERA-Interim's land-surface model still needs to be improved in terms of its simulation of soil hydrology and snow cover. This affects the utilization of satellite data over land surfaces within ERA-Interim, which has negative effects on the representation of clouds and precipitation (Berrisford et al., 2011). The confidence in temperature trend estimates, however, has improved considerably so that ERA-Interim data have been used as an alternative to observational datasets to monitor climate change (Willett et al., 2010).

### 2.2.2 Land use

We downloaded United States Geological Service (USGS) Land Use/Land Cover raster data from the global land cover characteristics database (U.S. Geological Survey, 2016). This was necessary, as early AVHRR data are distributed without masking information. The USGS data are used as a land sea mask within the optimal estimation retrieval, as well as a land cover classificator within the cloud mask and the Pavolonis cloud typing scheme. The dataset is defined on a regular lat/lon grid with 0.05° resolution. The USGS land cover classification was primarily derived from 1 km AVHRR Normalized Difference Vegetation Index (NDVI) 10-day composites for April 1992 through March 1993 (U.S. Geological Survey, 2016).

### 2.2.3 Land surface BRDF

MODIS Collection 6 Bidirectional Reflectance Distribution Function (BRDF) data (MCD43C1, Schaaf and Wang (2015)), providing kernel weights for the Ross-Thick/Li-Sparse-Reciprocal BRDF model, are used within the retrieval scheme to set surface albedo and bidirectional reflectance distribution conditions. These data are available every 8 days derived from cloud-cleared 16-day Terra and Aqua measurements, and provided in HDF-EOS format at 0.05° spatial resolution. MCD43C1 data are classified as high-quality given sufficient observations, and otherwise a low quality estimate is produced based on climatology anisotropy models. Validation against albedo measurements made at Baseline Surface Radiation Network (BSRN) sites show





that the black-sky and white-sky albedo computed from the single sensor MCD43A1 high-quality product are well within 5 % of the measured albedo, while the low-quality product is within 10 % (Lucht, 1998).

We regridded MCD43C1 data to instrument resolution through bilinear interpolation, and filled missing pixels within the time series with pixel values of the temporally closest 8-day composite file providing valid data. For the pre-MODIS era,

we produced a BRDF climatology by averaging all data available for a particular 8-day time slot. MCD43C1 kernel weights are applied to all CC4CL sensors, neglecting differences in spectral response functions as the surface is a relatively minor component of the observed signal.

### 2.2.4 Land surface emissivity

For land surface emissivity, we used the Cooperative Institute for Meteorological Satellite Studies (CIMSS) global land surface

infrared emissivity database created by the Baseline Fit method (Seemann et al., 2008). These data are derived from the MODIS operational land surface emissivity product (MOD11), to which the fit method is applied for filling spectral gaps between channels. CIMSS emissivity data are available on a monthly basis at ten wavelengths with 0.05° spatial resolution.

As for BRDF, we produced a land surface emissivity climatology for the pre-MODIS era by averaging all data available for a particular month.

### 2.3 Collocating CC4CL L2 data and CALIOP

We resampled CC4CL L2 data to a regular latitude/longitude grid at 0.1° × 0.1° resolution. This resampling is required for an intercomparison of CC4CL L2 data on a common grid, as differences in sensor spatial resolution are reduced when averaging all values available for each grid box. CALIOP's Level 2 5 km Cloud Layer data were produced by averaging over ∼14 beams with 70 m diameter taken every 335 m within a 5 km along-track corridor. Thus, CALIOP data have a 70 m across-track × 5 km

along-track spatial resolution (see also Holz et al. (2008)), and the size of the corresponding CC4CL grid box is approximately 11 km (meridional) × 2.9 to 5.6 km (zonal). As a consequence, the CC4CL grid boxes are larger than the reference CALIOP pixels, but are still small enough to resolve some of the cloud features that CALIOP observes. Note that AVHRR GAC data were produced by averaging 5 neighbouring pixels across-track, but CALIOP data were averaged along-track.

## 3 The CC4CL retrieval system

### 3.1 Heritage

In the early stages of the Cloud_cci project, a "Round Robin Exercise" evaluated three different algorithms regarding their applicability for retrieving cloud parameters from satellite data (Stengel et al., 2015), which were 1) the operational processing system of the CM SAF (2015), 2) the CLAVR-X algorithm used to generate the PATMOS-x climatology (Heidinger et al., 2013), and 3) the ORAC retrieval which was previously used to produce the GRAPE data set (Thomas et al., 2009b; Natural

Environment Research Council et al., 2015). All three algorithms were driven with identical MODIS and AVHRR input data





and ERA-Interim meteorological background information for five days in 2008. The results where analysed with respect to CloudSat, CALIOP and AMSR-E reference data.

Based on the outcomes of that study (Stengel et al., 2015), ORAC was selected to be the cloud retrieval scheme within CC4CL. Moreover, code modifications were identified and characterized to render ORAC fit for the purpose of ECV produc-
tion.

## 3.2 Preprocessing

The CC4CL preprocessor initially defines the dimensions and content of the sensor and preprocessing grids (Figure 01).

The sensor grid has the same extent and resolution as the input orbit or granule. The sensor grid is filled with sensor radiances and angles, time, and geolocation data (section 2.1), whereas surface BRDF (section 2.2.3), snow/ice coverage (from ERA-
Interim, section 2.2.1), and surface emissivity (section 2.2.4) are bilinearly interpolated onto that grid. We use BRDF data over land only. For sea pixels, the Cox and Munk ocean surface reflectance model calculates BRDF coefficients as a function of ERA-Interim wind speed. These coefficients also contain foam and underlight components (Sayer et al., 2010). The albedo of snow/ice covered pixels is set to globally constant values of 0.958 (Ch1), 0.868 (Ch2), 0.0364 (Ch3), and 0.0 (Ch4), and is area-weighted in the event of fractional sea/ice cover.

The preprocessing grid is a regular latitude/longitude grid that covers the extent of the sensor grid, but at a coarser resolution of $0.72° \times 0.72°$. It is used to store the average of all sensor angle and surface emissivity values falling within a grid box and spatially interpolated (nearest neighbour) land-use data (section 2.2.2). ERA-Interim variables were transformed before input to the preprocessing grid as described in section 2.2.1. For profile variables, vertical geopotential coordinates are calculated from pressure coordinates.

The preprocessor then calls the cloud mask (section 3.3.1) and cloud typing (section 3.3.2) algorithms. Finally, the Radiative Transfer for TOVS (RTTOV) model is executed on the preprocessing grid data as defined by ERA-Interim surface and profile variables. RTTOV outputs profiles of cloud transmittance both above and below cloud for the shortwave channels and emissivity for the longwave channels. For details on RTTOV and the forward model, see part II of this paper (McGarragh et al., 2017c).

All data are written to NetCDF files. In theory, the main processor would evaluate these inputs twice, assuming different cloud phases (e.g. ice and liquid). In practice, ORAC uses the preprocessed cloud mask and phase to select an appropriate method to reduce processing time.

## 3.3 CC4CL cloud retrieval

### 3.3.1 Cloud detection

The CC4CL cloud mask is produced by (1) estimating pseudo CALIOP cloud optical depth (ANNCOD) from L1 measurements with an artificial neural network (ANN), (2) correcting ANNCOD for viewing-angle dependencies, and (3) classifying ANNCOD into binary cloud mask information by thresholding.





CC4CL applies a set of ANN for cloud masking, one for each of the illumination conditions day (solar zenith angle $\theta_0 < 80°$), night ($\theta_0 \geq 90°$), and twilight ($80 \leq \theta_0 < 90°$). The ANNs are multilayer perceptrons with one input layer, one hidden layer with 50 neurons, and one output layer, which produces ANNCOD ranging from 0 to 1.

The various ANNs were trained with NOAA-18 AVHRR L1c data, auxiliary information (ECMWF land-sea mask, snow-ice mask, and surface temperature), and cloud optical depth (COD) "truth" data obtained from CALIOP'S 532 nm lidar product (CAL_LID_L2_05kmCLay-Prov-V3-01). AVHRR Ch3a data were generally excluded. We trained the day ANN with all remaining AVHRR channels, but also excluded Ch3b to be consistent with those NOAA platforms that switch between Ch3b transmission at night and Ch3a at day (NOAA-16, NOAA-17, MetopA) For night and twilight conditions, we produced ANNs both with and without Ch3b data input. This was necessary to avoid misclassification of very cold clouds and/or land surfaces due to Ch3b's very low signal-to-noise ratio. In addition to the days evaluated in the "Round Robin" comparison, we selected 12 further training days in 2008 that contain collocations between NOAA-18 and CALIOP, represent COD seasonality, and provide global coverage. Prior to training, all CALIOP COD values > 1 were set to unity. Auxiliary data input are the ERA-Interim skin temperature, a snow/ice mask derived from ERA-Interim snow depth and sea ice concentration, and the USGS land/sea mask. Finally, we applied a simple correction algorithm to remove a cosine viewing-angle dependency of retrieved ANNCOD. This was necessary, as the maximum viewing angle in the AVHRR training dataset was just 35°. The binary cloud mask is estimated by classification of ANNCOD data into clear and cloudy through a set of threshold values. The thresholds themselves vary depending on illumination and surface conditions, namely land, sea, and snow/ice cover (Table 02), and were quantified by trial and error. As the ANN was trained with AVHRR data only, differences in spectral response functions need to be considered before the ANN can be applied to MODIS and AATSR. We derived appropriate coefficients through linear regression analysis between collocated satellite observations for each input channel pair (Table 03). The resulting coefficients were applied to MODIS and AATSR satellite data before ANN input.

We estimate cloud mask uncertainty based on the assumption that this uncertainty is inversely proportional to the difference between retrieved ANNCOD and the threshold applied. As a first step, we generated a CALIOP cloud mask by application of a clear/cloudy threshold value of 0.05. The CALIOP cloud mask is then compared with the collocated ANN mask by quantification of a Percent Correct (PEC) score. PEC estimates the ratio between all correctly classified pixels and the number of all pixels analysed. Finally, the "truth" uncertainty is defined as $100 - $ PEC %. We then established the statistical relationship between this uncertainty and the ANNCOD difference to its threshold. Before application of the approach, we normalised differences (ND) to 1. We found a linear correlation between uncertainty and ND for clear cases given by

$$y = 37.275 \times \text{ND} + 49.2, \tag{1}$$

and a second order polynomial correlation for cloudy cases (Figure 02)

$$y = 54.133 \times (\text{ND} - 1)^2 + 1.862. \tag{2}$$

The equations of these regression fits are used within CC4CL to quantify cloud mask uncertainty as a function of ND.



### 3.3.2 Cloud typing

Cloud phase is determined by application of the Pavolonis cloud typing algorithm (Pavolonis et al., 2005). The Pavolonis algorithm outputs 6 cloud types (Table 04), which we then reclassified into water or ice clouds: liquid = fog/warm liquid/supercooled, ice = opaque ice/cirrus/overlap. For CC4CL, the fog type test was deactivated. The algorithm always uses

the 0.65, 11, and 12 μm channel data. It reads 3.75 μm data whenever available, and 1.65 μm otherwise. These two different approaches produce nearly identical results, except for certain thin clouds and cloud edges (Pavolonis et al., 2005). In addition, we introduced two new cloud types within CC4CL. In response to validation studies, we decided to change the phase of ice clouds whose retrieved CTT is > 273.16 K, the freezing point of water (new cloud type = SWITCHED_TO_WATER), and of water clouds whose CTT < 233.16 K, the lower limit of supercooled water (SWITCHED_TO_ICE).

The Pavolonis algorithm has weaknesses in detecting cirrus clouds at high latitudes, which are often misclassified as opaque ice clouds. Performance is considerably better when the VIIRS algorithm is used, which provides additional channels and threshold tests. However, these cannot be applied to our AVHRR heritage dataset (Pavolonis et al., 2005).

### 3.3.3 Optimal estimation retrieval of COT, CER and CTP

The optimal estimation retrieval ORAC is a non-linear statistical inversion method based on Bayes' theorem (Rodgers, 2009).

A state vector containing all variables to be retrieved is optimized to obtain the best fit between observed TOA radiances and radiances simulated by a forward model. The retrieval problem is that of finding the minimum value of a cost function. This function is based on a $\chi^2$ distribution, which is a combination of the squared deviations between the measurements and the forward model and the retrieved state vector and the a priori state vector, each weighted by their associated uncertainties. The important benefits of ORAC, relative to more traditional retrieval methods, are that cloud parameters are retrieved using

information in all satellite channels simultaneously, so that the retrieved parameters provide a robust representation of the short-wave and long-wave radiance effects of the observed cloud. The algorithm estimates the retrieval uncertainty, which can be thought as a measure of the consistency between the retrieved cloud parameters and the satellite measurements (Poulsen et al., 2012). For a more detailed description of the ORAC algorithm see part II of this publication (McGarragh et al., 2017c).

### 3.4 Post-processing

For each input pixel, the main processor produces retrieval values for both ice and liquid clouds. The postprocessor will then select the appropriate output variables according to the Pavolonis cloud phase. As described in section 3.3.2, the postprocessor changes cloud phase in case retrieved CTT does not match the Pavolonis phase. Finally, output variables are written to primary and secondary NetCDF files (Table 05).



## 4 L2 data - analysis and initial validation

We first examine CC4CL cloud properties for one sample scene that extends from approximately 100° W to 170° W and 45° N to 75° N over North America. We focus on the consistency of retrieval values derived from different sensors (AVHRR, MODIS, AATSR). This includes pixel-based uncertainties of the key variables (CTP, COT, CER, and cloud mask). We then
perform a validation of retrieved cloud properties, for which CALIPSO data are our reference. This validation is limited to three high-latitude scenes for which collocations for all sensors with CALIOP are available.

### 4.1 CC4CL cloud properties

The sample scene is characterized by various cloud types, and the CC4CL cloud mask defines a relatively small fraction as cloud free (Figures 03 to 05). Visually, similar spatial patterns are observed in the three products. The data show that there are
more cloud free AVHRR pixels, which is related to the coarser spatial resolution compared to MODIS and AATSR. The LST difference is $\leq$ 5 minutes, so there is little cloud displacement between observations.

    CTP data are approximately normally distributed for all three sensors. Both COT and CER show positive kurtosis and skewness, as values close to 0 are common. CER data are somewhat bimodal, having a primary peak at ∼12 μm and a secondary peak at ∼35 μm (Figure 07 and Table 06). Mean value differences are not significant between AVHRR and MODIS for CTP,
MODIS and AATSR for COT, and AVHRR and AATSR for CER. The standard deviation of differences between two sensors are always lowest for AVHRR minus MODIS (Table 06). Significance tests of mean differences and standard deviations of residuals between sensor retrievals are sensitive to outliers, and are to some extent influenced by cloud displacement due to observation time differences. Even though we found no significant relationship between sensor retrieval residuals and observation time difference (not shown), residuals are likely to be smaller and thus possibly insignificant if sensor observation times
were identical.

### 4.2 Uncertainties

Median absolute uncertainties are CTP = 26.7 hPa, COT = 6.1, CER = 2.0 μm, and cloud mask = 13.7 % (Figure 06). The median relative retrieval uncertainty (not shown) is relatively low for all three retrieval variables (CTP = 4.7 %, COT = 6.1 %, CER = 2.0 %). COT uncertainties increase with COT magnitude, and the RGB image (Figure 010) shows that the largest
uncertainties are found in cases of opaque cloud coverage and cloud over sea-ice surfaces. CER results are similar to COT, although relative uncertainties are somewhat lower. Cloud-free areas show increased cloud mask uncertainties, particularly over sea-ice surface areas. Note that the cloud mask uncertainties have been quantified as a function of the normalized difference to the cloud mask threshold, whereas relative retrieval uncertainties (100 × uncertainty ÷ retrieved value) are shown for CTP, COT, and CER.



## 4.3 Validation with CALIOP

We found collocations between CALIOP, AVHRR, MODIS, and AATSR for three study areas in the Arctic at 07/22/2008 19:15 LST (study area North America 1 = NA1, n = 120, Figure 08), 07/22/2008 20:58 LST (NA2, n = 163, Figure 010), and 07/27/2008 08:10 LST (Siberia = SIB, n = 116, Figure 012). These are located within 60° to 75° N latitude, and contain

vegetated land, snow-covered land, open ocean, and sea-ice surfaces. For NA1 and SIB, all CALIOP pixels were classified as cloud covered, while for NA2 about half of the pixels are cloud free.

When including AATSR, collocations are restricted to high latitude areas and by the narrow swath of AATSR. We thus decided to include another scene without AATSR data in the Gulf of Guinea/West Africa between 7° S and 12° N at 24/10/2009 13:45 LST (Africa = AFR, n = 1181, Figure 014). There, about ten times more pixels are available than in the other scenes and

cloud systems not contained in the Arctic data are observed, such as low-level stratocumulus and deep convection.

We divided all study areas into logical sectors, for each of which a characteristic pattern of cloud coverage and type predominates. The validation is shown for comparisons of CTH rather than the retrieved value to enable a more intuitive visualization and discussion. CTH is derived using the retrieval's atmospheric profile. An important caveat to note is the difference between physical and radiating cloud top. CALIOP uses an active sensor that is (roughly) sensitive to particle number. It identifies

what we call the physical cloud top, denoted by the sharp increase in particle number. The passive radiometers analysed using CC4CL are (roughly) sensitive to the temperature of the cloud, from which the height is calculated. However, TOA radiation is the sum total of emission and scattering throughout the atmospheric column observed. As there is no single height contributing, the retrieved CTT is more accurately described as an effective radiating pressure, being an average of the cloud's temperature profile weighted by the probability that a photon from each level can arrive at the detector. As a rule of thumb, the observed

CTT represents the state one optical depth into the cloud. For the purposes of comparison to CALIOP, CC4CL computes a 'corrected' CTH, which adjusts the retrieved CTH to where the physical cloud top would be expected, assuming an adiabatic profile.

### 4.3.1 Case studies

*Case study NA1*

Study area NA1 is a completely cloud-covered scene over northern Canada containing clear and ice-covered land and open ocean surfaces. There are a variety of single and multi-layered clouds. CC4CL correctly classifies all pixels as cloud covered, with a few exceptions in sectors 3 and 4. CTH retrievals are consistent between the three sensors, only differing in sector 2. CTH is generally lower than CALIOP's top layer height, unless the latter is optically thick as in sector 4. In the case of a (semi-)transparent cloud top layer, multiple surfaces contribute to the observed satellite data. CC4CL CTH is then located

closer to, at, or even below the underlying cloud layer (sectors 1, 3 and 2, respectively). For single-layer, optically thick (COT > 1) cloud, CC4CL and CALIOP CTH agree very well (sector 4). Under such conditions, the retrieval is very accurate. Cloud





phase agreement between CC4CL and CALIOP is very variable. It is best for optically thick high ice cloud coverage (sector 1), and worst for low water clouds (sector 4).

### Case study NA2

Study area NA2 is located entirely over snow/ice free land in Western Canada. CALIOP cloud coverage is 4.5 %, spatially broken, and variable in height and phase. Clear-sky pixels are mostly identified by CC4CL (69.3 % correct), and cloudy pixels are occasionally missed (78.7 % correct). CC4CL retrievals of thin high clouds and false positive cloudy pixels have low CTH values (sector 1). Small-scale horizontal variability in CALIOP cloud phase is reflected by CC4CL data, which overestimate the fraction of liquid water clouds in sector 2. CC4CL reproduces CALIOP's spatial variability in CTH, which it slightly underestimates by 0.5–1 km in sector 2. In sector 3 CC4CL considerably underestimates CTH by up to 7 km. Most of these clouds are optically and geometrically thin.

### Case study SIB

Study area SIB crosses the Novaya Zemlya islands north of Siberia and is defined by a mixture of open ocean and partially snow/ice covered land surfaces. According to both CALIOP and CC4CL, it is completely cloud covered.

In the event of single-layer cloudiness, CC4CL CTH agrees very well with CALIOP (sector 1 and, in particular, sector 3). The CTH difference between CC4CL retrievals increases in the presence of overlapping clouds (sector 2). There are optically thin but vertically thick (∼4 km) clouds in sector 2. For these the retrieved CTH is considerably underestimated by ∼6 km, which is probably a result of lower layer contributions that "contaminate" the satellite signals. Overall, about 62.3 % of CC4CL pixels agree with CALIOP phase. Phase mismatch occurs in cases of single layer optically thin clouds (sector 2) and, less frequently, stratiform cloudiness (sector 3).

### Case study AFR

Study area AFR is located over the Gulf of Guinea and Western Africa, containing open ocean and snow/ice free land surfaces. As we excluded AATSR data, about 10 times more pixel collocations with CALIOP are available (n = 1181) than for previous cases. Additionally, measurements contain tropical and coastal cloud systems such as extensive low-level stratiform cloudiness and continental deep convection (Figure 014).

In general, the quantitative and qualitative agreement between CC4CL and CALIOP CTH is impressive, compared to the performance of existing algorithms. CC4CL data track the spatial pattern of continental CTH very well (sector 3), which increases northwards and shows some small scale variability beyond 8° N. However, CC4CL underestimates CTH of vertically thick clouds and instead places the cloud top at a layer's vertical centre. The height of the stratiform cloud field is almost identical for CC4CL and CALIOP, although CTH of near-surface stratiform clouds is overestimated below thin high cirrus (sector 1). For the small layer located at 4 km height at ∼ 6.7° S and the thin high cirrus layer around 2° S, MODIS retrieval values differ somewhat from CC4CL AVHRR and CALIOP. Again, the phase of optically thick clouds is retrieved very well,



which however is not the case for the thin ice cirrus clouds. Generally, CC4CL using the AVHRR heritage channel dataset are almost entirely insensitive to the very high, thin cloud layer in sector 1 (covering stratiform clouds) and 2 (covering the sea surface). Here, CC4CL is rather driven by contributions from very low clouds or the sea surface.

### 4.3.2 Validation summary

The four study areas clearly show that CC4CL retrievals of CTH are very close to CALIOP values for single layer, optically thick clouds. For significant extents of the regions presented, the CTH is accurate to within 240 m. For multi-layer clouds, CC4CL estimates are almost exclusively located in between CALIOP's top and bottom layer estimates. For these cases, the optimal estimation algorithm processes satellite signals that are likely to contain radiance contributions from multiple cloud layers. The OE then optimizes the fit between modelled and observed radiances by placing the cloud lower in the atmospheric profile, and so the mixed nature of the satellite data leads to an underestimation of CTH. The results shown here are a representative sample from an extensive validation performed within the Cloud_cci project..

There is no clear influence of the underlying land type or topography on retrieval values or the cloud mask. However, the limited sample size does not allow for generalizations. For site NA2, CALIOP identified cloud-free pixels, 69.3 % of which were also detected as cloud-free by CC4CL's neural network cloud mask, and with few exceptions as low level water clouds otherwise. In relatively few cases, CC4CL fails to detect clouds seen by CALIOP (% of missed clouds = 9.0 (NA1), 21.3 (NA2), 0.6 (SIB), 3.1 (AFR)). We did not account for fractional cloud coverage, as we set a grid box as cloud covered if any corresponding CC4CL pixel contains cloud information. As a consequence, there are slightly more cloud covered pixels for the spatially higher resolved MODIS and AATSR data than AVHRR.

The CC4CL phase identification agrees continually with none of the three CALIOP cloud flags shown, which is sensible given the differences between active and passive observations. After rounding CC4CL values to the nearest integer, the percentage of pixels with equal phase is lowest for the top layer at COD > 0 (NA1 = 49.9 %, NA2 = 32.2 %, SIB = 62.3 %, AFR = 53.0 %), but similar for the mid layer COD > 0.15 (NA1 = 53.1 %, NA2 = 46.5 %, SIB = 66.4 %, AFR = 94.7 %), and the bottom layer COD > 1 (NA1 = 47.3 %, NA2 = 57.4 %, SIB = 66.3 %, AFR = 91.1 %). These values however do show that phase determination performs very well if optically thick clouds dominate, as is the case for study area AFR. When averaged over all layers, phase agreement is largest for site AFR (79.6 %), followed by SIB (65.0 %), and clearly lower for NA1 (50.1 %) and NA2 (45.4 %).

For ice clouds, the most frequently occurring cloud types are cirrus (ID=6) for CALIOP and overlap (ID=8) or cirrus (ID=7) for CC4CL. Water cloud types are more heterogeneous and for CALIOP predominantly low transparent (ID=0), but altostratus (ID=5) and altocumulus (ID=4) are also frequent. CC4CL water clouds are approximately equally distributed amongst water (ID=3) and supercooled (ID=4) cloud types.

The scenes investigated and discussed here are just a small subset of the large variety of global cloudiness. We could only find collocations with AATSR data at high latitudes, where multi-layer cloud coverage is common. Under such difficult conditions, the retrieved CTH is a mixture of all radiatively contributing cloud layers. Please refer to Stengel et al. (2017) for a quantitative, global validation of CC4CL cloud properties.



## 5 Discussion

### 5.1 The flexibility of the optimal estimation approach

In general, the retrieved values are insensitive to the specific instrument evaluated, such that the merged data set is sensible..
Absolute mean differences are $\leq$ 21.9 hPa for CTP, $\leq$ 1.3 for COT, and $\leq$ 2.1 for CER. These are mostly smaller than the mean
retrieval uncertainties themselves. Moreover, the RGB images show that all major patterns of cloud coverage and structure are
resolved by all three sensors. However, AATSR data show larger deviations than the other sensors (Figure 07). It is unlikely that
differences in spectral response functions are the reason. MODIS and AATSR heritage channels are relatively close in their
spectral response but their retrieval values do differ considerably. Also, MODIS and AVHRR disagree nonetheless more in
their spectral response, which results in a reflectance difference of up to 30–40 % (Trishchenko et al., 2002), but their retrieval
values are much more similar. The difference to AVHRR and MODIS is largest for CER, so microphysical variables, which
are derived from reflectance data only, appear to be most affected.

The differences between mean values are almost always significant (t-Test, $H_0$: $\mu1 = \mu2$). Thus, from a statistical point of
view, the samples we analysed for AVHRR, MODIS, and AATSR have been drawn from different populations and are thus
statistically inconsistent. In other words, the retrieval system should not produce statistically consistent cloud parameters when
driven with satellite data obtained from three spatiotemporally collocated sensors. However, differences in cloud conditions
at the various observation times and sensor spatial resolution explain part of these discrepancies. Moreover, a non-significant
t-Test result is possibly too strict a metric for estimating the consistency of retrieval results. There is a range of confounding
processes that affect each individual retrieval estimate, such as observation times, spectral responses, calibration deficiencies,
and a varying number of cloudy pixels to be compared. The case studies clearly show that, under optimal conditions for single
layer cloud retrievals, CC4CL products are consistent with CALIOP and practically insensitive to sensor characteristics.

We suggest that AVHRR and MODIS data can be used interchangeably, depending on the user's application. AVHRR data
provide long-term data records from 1982, however at a relatively coarse resolution of 5 km × 3 km. The MODIS data record
started in 2000, and is thus too short to be considered a climatology. However, L1 data are available at 1 km resolution, and
orbit control is guaranteed. With CC4CL, we also produced 0.05° lat/lon daily composites for Europe (data not shown), which
is close to MODIS's original resolution in that area. These data provide a more detailed view on cloud features than AVHRR.
In that sense, CC4CL products retrieved from AVHRR and MODIS are complementary. More detailed analysis is required to
assess differences in CC4CL output data when applied to AATSR data.

### 5.2 The value of uncertainty quantification

The retrieval uncertainties prove to be a valuable source of information. On the one hand, they are useful for several user
applications, such as model validation, data assimilation applications, or climate studies in general. On the other hand, they
allow for diagnosis of potential retrieval shortcomings. For example, we see that COT uncertainty scales with COT itself and is
thus heteroscedastic (see also Poulsen et al. (2012)). CC4CL COT values are at times unnaturally large, and the associated un-
certainty reflects that. Also, it highlights under which conditions the optimal estimator converges to a solution with a relatively



large divergence from the measurements. In the cases shown here, large uncertainties are associated with optically thick clouds or underlying snow/ice cover. COT and CER uncertainties are clearly largest, and reflect the limited information available with which to retrieve these values. For further possible explanations due to assumptions and limitations within the methodology applied, please see part II.

We applied an independent approach to quantify cloud mask uncertainty. It is valuable information, as a neural network does not provide output uncertainty. The approach we adopted here is straightforward. When the NN output, which is a pseudo CALIOP COD, approaches a defined threshold value for cloudiness, the uncertainty increases towards a maximum of 50 %. This maximum value indicates that a cloud mask value is basically random, as it is equally likely to be cloudy or cloud-free. With that in mind, the cloud mask uncertainty data are easy to interpret. For example, we see that sea-ice pixels classified as

cloudy to the North of study area NA2 (Figure 06) show uncertainties of 40 - 50 %. This indicates that the NN is sensitive to bright ground cover, which may be confused with clouds. We suggest that users of ESA Cloud_cci data should routinely consult cloud mask uncertainties. If a more conservative cloud mask is required, it can be easily built by setting a maximum value for an acceptable uncertainty level.

### 5.3   Strengths and weaknesses

The results clearly show that CC4CL retrieves CTH of single layer, optically thick clouds with high accuracy and precision. When compared to CALIOP, the mean deviation for these cases is as low as 10–240 m CTH. This is a promising result, and shows that the optimal estimation framework is robust and appropriate for retrieving cloud properties from AVHRR heritage channels.

    In the case of multi-layer clouds, CC4CL is not able to retrieve CTH of the top layer if it is optically thin. The estimate

is somewhere between the two cloud pressure values. This is an expected limitation of our framework, and also of other retrieval algorithms using passive sensor data (Holz et al., 2008; Karlsson and Dybbroe, 2010). Poulsen et al. (2012) found that ORAC CTP and CER estimates are robust when the top ice cloud layer is > 5 optical depths, and otherwise are the weighted average of several cloud layers. The AVHRR heritage channels do not provide sufficient information on retrieving cloud vertical structure. In the case of semi-transparent top-layer clouds, the upwelling signal, may it stem from a cloud or the

Earth's surface, contributes to the total TOA reflectance or brightness temperature. This mixed satellite measurement is input to CC4CL, which retrieves cloud parameters assuming a single cloud layer. As brightness temperatures and reflectances of, e.g., a cold, semi-transparent, bright top-layer cirrus cloud overlapping a warmer, opaque, and darker low-level water cloud, are a mix of several contributing surfaces, so will be the final retrieval value. Any CTH retrieved from AVHRR (heritage) data is the radiatively effective than physical cloud top (Karlsson et al., 2013). For CC4CL, we often see that the final CTH estimate

is placed between top and lower levels and is thus an underestimate, which is a common problem amongst retrieval algorithms using passive sensor data (Watts et al., 2011; Holz et al., 2008; Karlsson et al., 2013).

    Multi-layer cloud property retrievals have been developed (Watts et al., 2011), and we also implemented and tested such an approach within CC4CL (McGarragh et al., 2017b). However, this method requires MODIS channels beyond the AVHRR heritage set, and thus will not be applicable to a full AVHRR reprocessing. For ESA Cloud_cci, the decision was made to



deliberately trade spectral information for time series continuity. Thus, discontinuities due to changing spectral coverage within the entire dataset are avoided (Stengel et al., 2017). In addition, we introduced corrected estimates of CTP and the derived CTT and CTH to get closer to the physical or geometrical cloud top. The correction is based on a vertical displacement of CTP along the atmospheric profile based on optical thickness and the cloud's extinction coefficient, which is a function of CER

(McGarragh et al., 2017a). The correction is only made for ice phase clouds.

On a first view, estimates of cloud phase appear reasonable when compared to CALIOP. However, we find the best overall agreement of $\sim 65\,\%$ for the lower layers (cumulative COD $> 0.15$ or $> 1$). This is just slightly better than a random guess of cloud phase. Cloud phase is generally difficult to quantify, and estimates of various satellite derived products disagree considerably for that variable (Stengel et al., 2015). An evaluation of MODIS Collection 6 cloud phase yielded a total cloud

phase agreement of over $90\,\%$ with CALIOP. However, as the study is exclusively based on single-phase cloudy pixels, the performance of MODIS C6 as applied to multi-phase pixels is still unknown (Marchant et al., 2016). We also find very high scores for cloud phase determination if restricting the analysis to optically thick, spatially extensive cloud fields such as in study site AFR. There, cloud phase agrees with lower layer CALIOP estimates by as much as $95\,\%$.

The key problems for phase determination are vertical stratification and the lack of direct in-situ measurements of cloud

phase. CALIOP observations, and also DARDAR (radar lidar, Ceccaldi et al. (2013)), are currently considered to be the most advanced estimates of cloud phase, relying on active measurement principles with depolarization and total attenuated backscatter at multiple wavelengths for additional constraints (Winker et al., 2009; Karlsson and Dybbroe, 2010). However, this assumption is primarily based on the physical theory underlying their retrievals, rather than on a comprehensive validation with independent observations of cloud phase.

Within CC4CL, we apply the Pavolonis algorithm for phase detection (Pavolonis et al., 2005). It was designed using simulated radiance data for varying phase, and further adjusted after analysis with real satellite data. The algorithm itself is a decision tree that contains a set of fixed threshold values for input reflectances and brightness temperatures, and was tuned to AVHRR. Even though we expect differences in phase determination between AVHRR vs. MODIS and AATSR due to varying spectral response functions, these were not large for the three study sites. Pavolonis et al. (2005) state that their product

could not be validated due to the lack of direct observations, but rather underwent a consistency check with ground-based, independent estimates.

The relatively low degree of agreement between CC4CL and CALIOP is not satisfying if CALIOP is considered to be the truth. However, we refrain from concluding that the CC4CL phase estimate was unrealistic as, to date, no robust, spatially resolved in-situ observations are available and our comparisons included multi-layered cloud conditions. It is difficult to de-

30 termine the representative CALIOP cloud layer when validating a passive sensor retrieval. For single layer, optically thick clouds, CC4CL can be compared with any layer exceeding a cumulative optical thickness of 0 or 1. If such a cloud layer was covered by optically and geometrically thin cirrus clouds, the satellite data are still dominated by lower cloud level reflectance and, in particular, emittance. Consequently, Pavolonis cloud phase is not a top layer estimate in such cases. For study area AFR, we also found situations where the NN cloud mask, which was trained with CALIOP data, correctly identifies thin high

cirrus as cloud over ocean but the cloud type algorithm failed to identify its ice phase. One potential improvement would be to



use the NN to provide an estimate of cloud phase. Initial tests indicated that this approach would indeed improve the (global) agreement with CALIOP, which is to be expected, as the NN is trained with CALIOP data. However, no estimate of cloud type would be provided.

CALIOP data are considered to be the current benchmark of cloud detection, vertical structure, and phase (Winker et al., 2009; Karlsson and Johansson, 2013; Holz et al., 2008), and are – except for the cloud mask – a source of validation with absolute independence from CC4CL. The main limitation of CALIOP though is its narrow view, so that global coverage is very limited. Also, the instrument is only able to probe the full geometrical depth of clouds whose total optical thickness is not larger than about 3–5 (Karlsson and Johansson, 2013). We found no clear relationship between CC4CL CTP uncertainty and the difference between CC4CL CTP and CALIOP CTP (data not shown). This suggests that the AVHRR heritage channels provide independent information on cloud vertical structure that is not clearly related to CALIOP's CTP estimates. Retrieval uncertainty is driven by background and forward model error as well as the mixed-signal satellite radiances rather than by the complex, real vertical cloud structure.

## 6   Conclusions

We have shown that CC4CL is a robust and flexible framework for producing cloud products from passive satellite sensor data. Differences between retrieved values for collocated satellite data are smaller than estimated uncertainties for AVHRR, MODIS, and AATSR. ESA Cloud_cci data provide climatologies (AVHRR) as well as highly resolved snap-shots for selected regions (e.g. Europe, MODIS). The complete sensor set of CC4CL data forms a unique, coherent, long-term, multi-instrument cloud property product that exploits synergic capabilities of several EO missions. Compared to single sensor retrievals, CC4CL data are improved in terms of accuracy and spatiotemporal sampling.

CC4CL explicitly estimates retrieval uncertainties according to the principles of error propagation through optimal estimation theory. These uncertainties are a valuable source for model validation, data assimilation, climate studies, or retrieval diagnosis. Cloud mask uncertainty is a novel feature that enables the user to assess product quality and to create individualized cloud masks.

We find that CC4CL is limited by weaknesses that are common to passive sensor cloud product retrievals. In general, an initial validation with CALIOP data shows that the CTH of optically thin clouds is underestimated. In the case of multi-layer clouds, the retrieved CTH is a mixture of all radiatively contributing cloud layers. The AVHRR heritage channels do not provide sufficient physical information that would allow for detailed retrievals of cloud vertical structure. Moreover, the forward cloud model is structurally incomplete, as it assumes a single-plane cloud layer. A multi-layer cloud property retrieval has been added to CC4CL, but is only applicable to MODIS data.

To account for CTH underestimation, we implemented a correction for CTH that assumes that passive sensor data see beyond the top into the clouds up to a penetration depth of $\sim 1$ optical depth. Corrected cloud top values are stored as separate variables within CC4CL output files.



Similarly, we find that the cloud phase estimate is only accurate for optimal retrieval conditions (optically thick top clouds). In a subsequent reprocessing of the AVHRR data record, we replaced the Pavolonis et al. (2005) algorithm with a neural network cloud phase estimation with better performance scores.

Under optimal conditions for single layer cloud retrievals, CC4CL products show little sensitivity to sensor characteristics. Single layer, optically thick cloud retrievals are very accurate and precise when compared against CALIOP (bias $<$ 240 m), which emphasizes the maturity and robustness of CC4CL. We thus recommend ESA Cloud_cci data to be used for multi-annual studies of cloud parameters and more detailed assessments of regional patterns and diurnal variability.

*Acknowledgements.* NASA kindly provided MODIS Collection 6 radiance data. ECMWF kindly provided support and assistance with their computing facilities for development and processing. This research/work was supported by the European Space Agency through the Cloud_cci project (contract No.: 4000109870/13/I-NB).



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





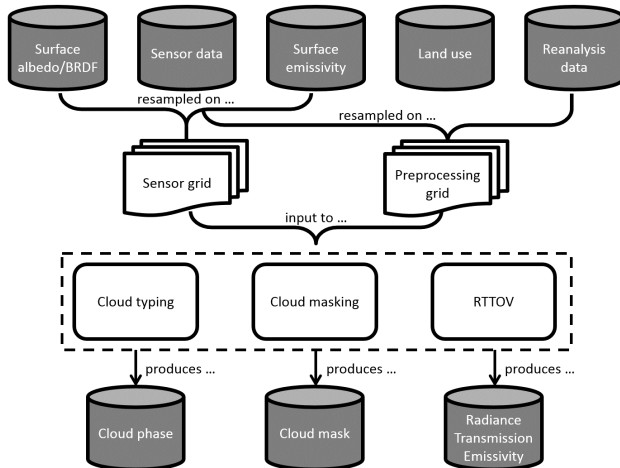

**Figure 1.** Schematic of the CC4CL preprocessor.





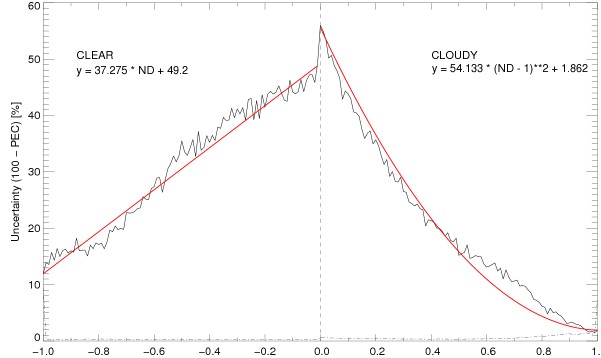

**Figure 2.** Neural network cloud mask uncertainty as derived from observations.




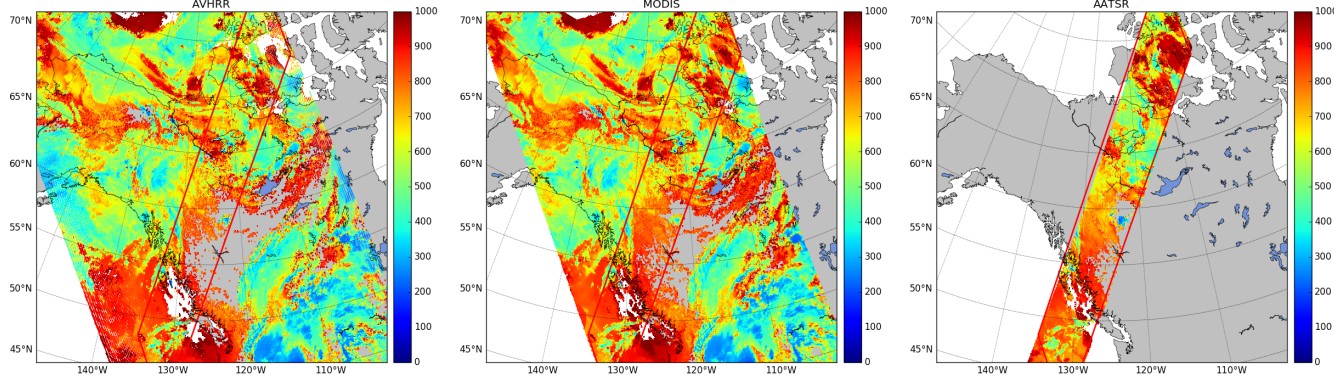

**Figure 3.** CTP retrieval values for study area NA2 with data from AVHRR (left), MODIS (middle), and AATSR (right).

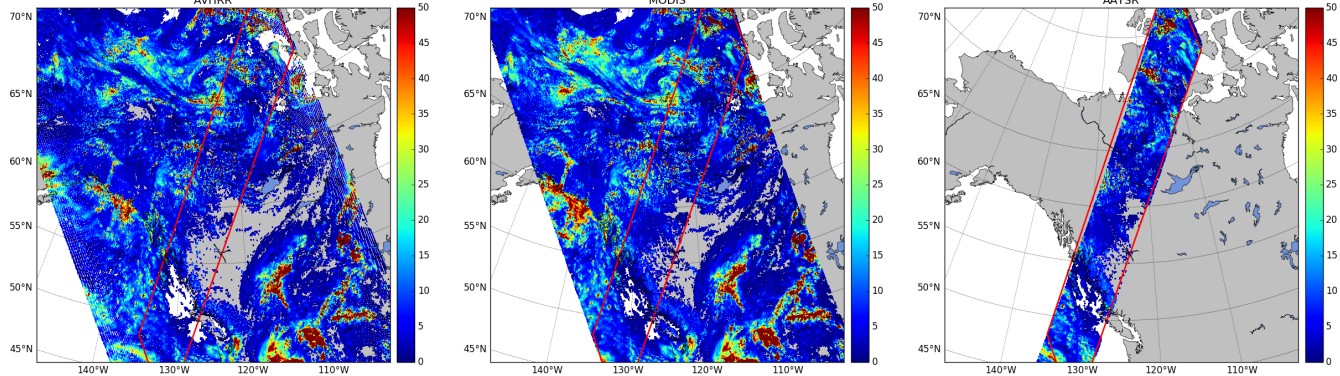

**Figure 4.** COT retrieval values for study area NA2 with data from AVHRR (left), MODIS (middle), and AATSR (right).

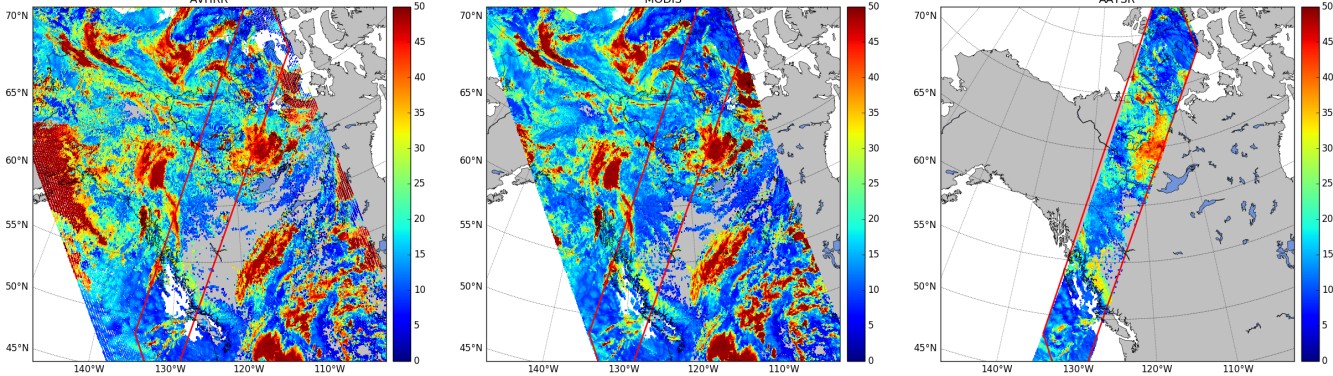

**Figure 5.** CER retrieval values for study area NA2 with data from AVHRR (left), MODIS (middle), and AATSR (right).





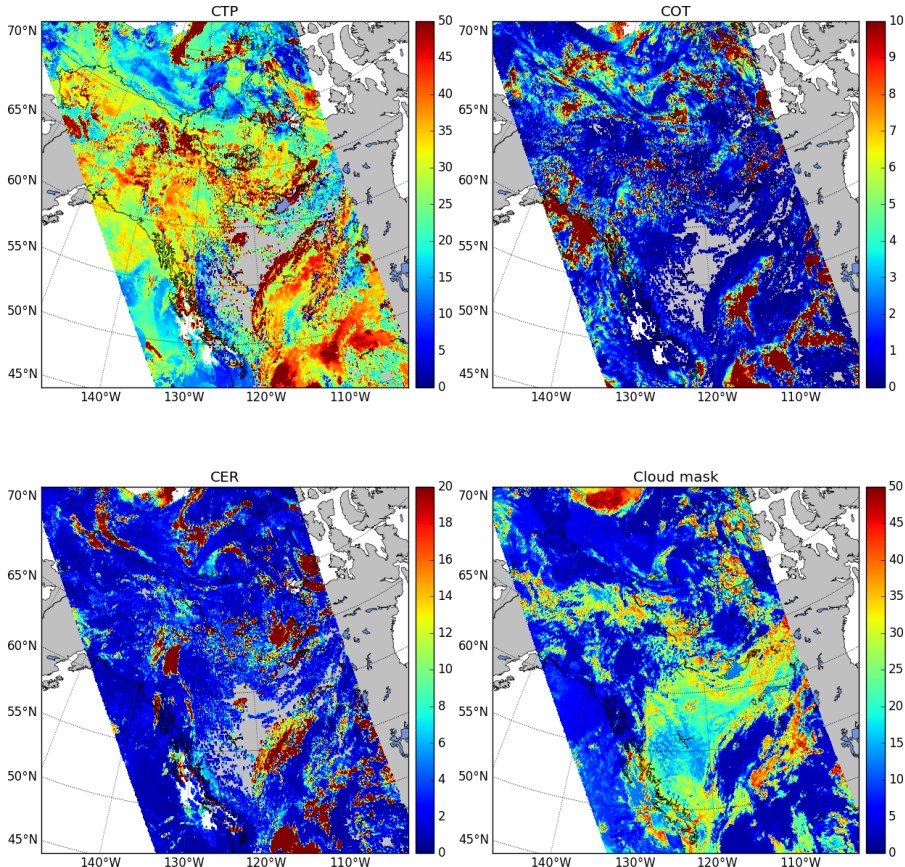

**Figure 6.** Absolute uncertainties of MODIS AQUA retrieval data for study area NA2 and CTP [hPa], COT, CER [μm], and Cloud mask [%].



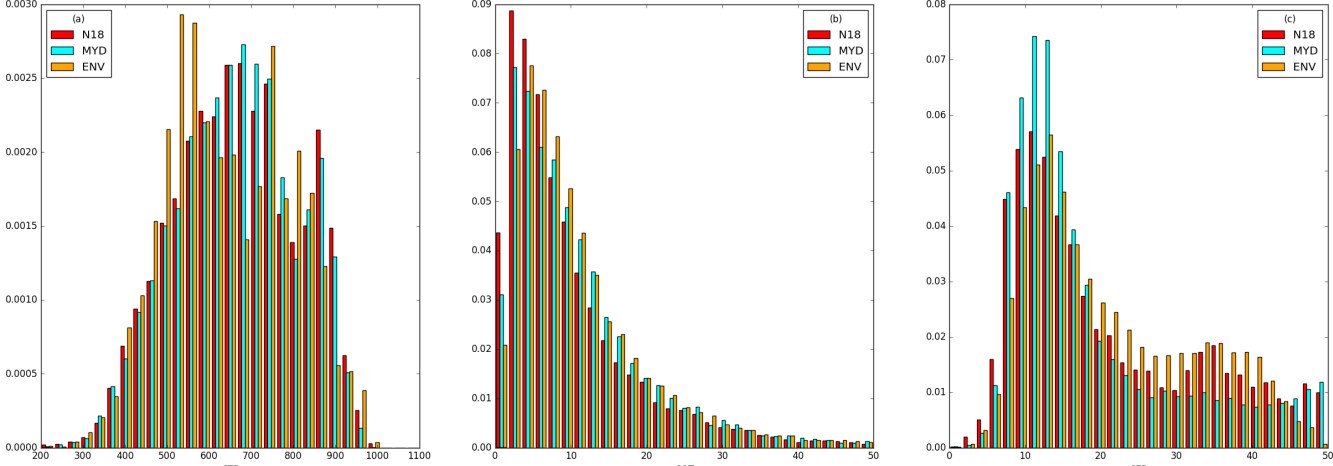

**Figure 7.** Density histograms of NOAA18 (N18), MODIS AQUA (MYD), and AATSR (ENV) retrieval data for study area NA2 and (a) CTP, (b) COT, and (c) CER.





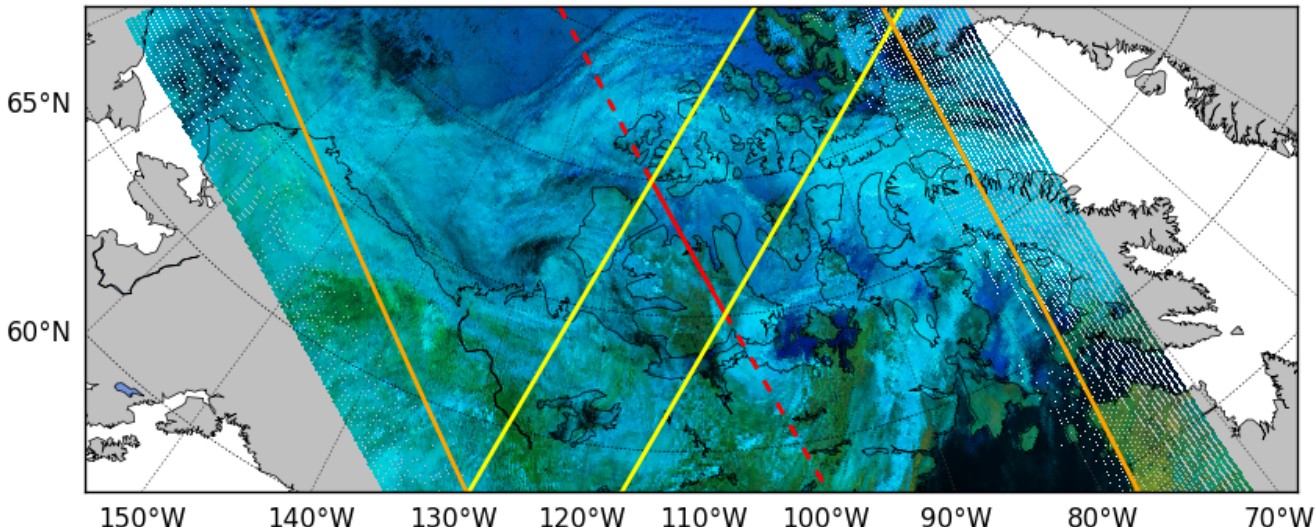

**Figure 8.** Study area NA1 (North America 1). Red (Ch1), green (Ch2), blue (Ch4 - Ch5) image derived from NOAA18 data resampled to 0.01°×0.01° resolution. Date of observation is 07/22/2008, 19:15 LST. Orange lines: extent of the collocated MODIS granule, yellow lines: extent of the collocated AATSR orbit, red line: CALIOP track outside (dashed) and within (solid) study area.

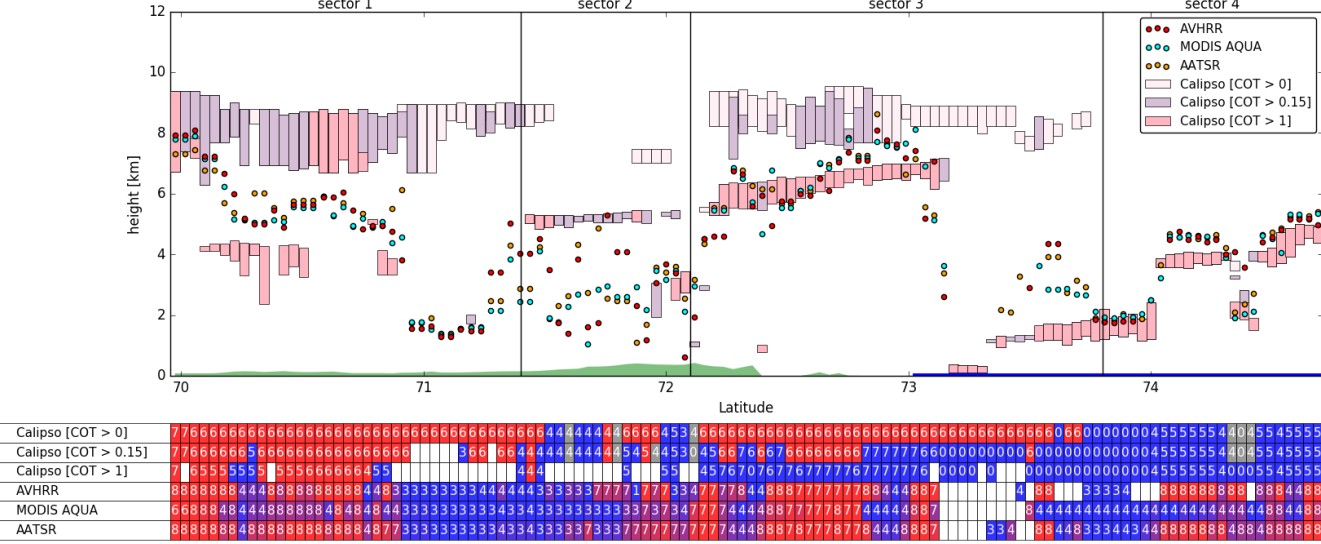

**Figure 9.** Vertical cross section of study area NA1 (North America 1) along the CALIOP track at 5 km horizontal resolution. Top: CTH for CC4CL retrievals (coloured points) and CALIOP measurements (vertical bars), and surface elevation and surface type (blue = open water, green = land, grey = snow/ice). The CALIOP data are shown for those pressure layers where the cumulative top-to-bottom COD exceeds a threshold value of 0 (top layer), 0.15 (mid layer), and 1 (bottom layer). Bottom: Cloud mask/phase (ice to water = red to blue, cloud free = white, not determined = grey) and type (see Table 04 for key/value pairs) for all three CALIOP layers and CC4CL retrievals. For CC4CL, cloud phase was averaged when resampling, and cloud type was assigned to the most frequent class per grid box. Sectors of characteristic cloud fields are separated by black vertical lines. Number of pixels n = 120



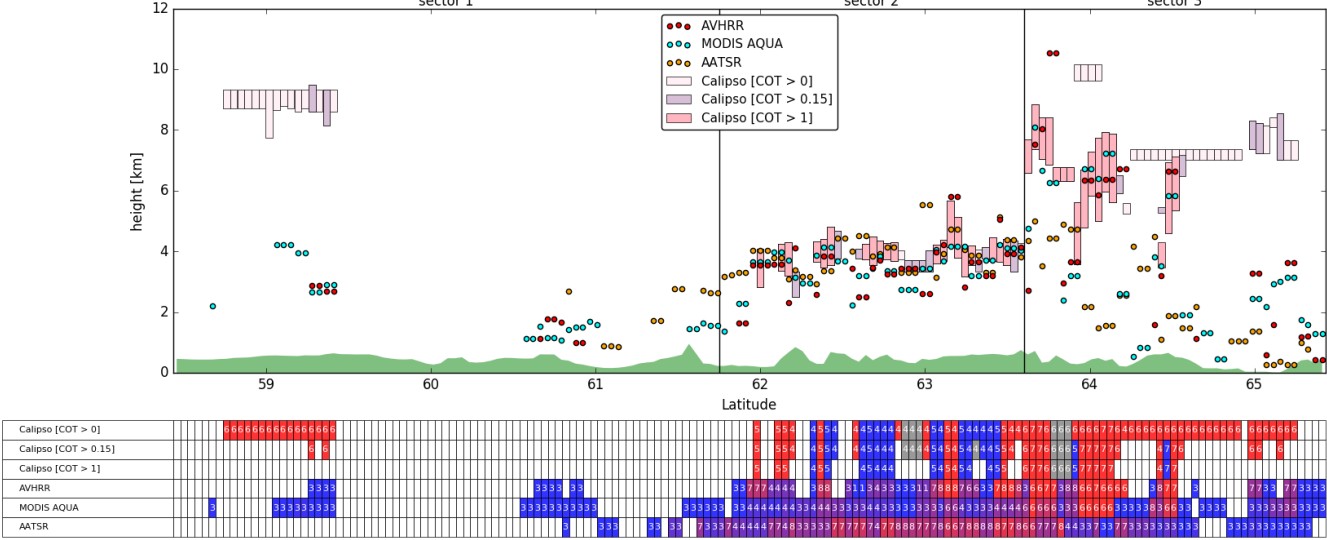

**Figure 10.** Study area NA2 (North America 2). As Figure 08, but at 07/22/2008, 20:58 LST.

**Figure 11.** Study area NA2 (North America 2). As Figure 09, but at 07/22/2008, 20:58 LST (n = 163).



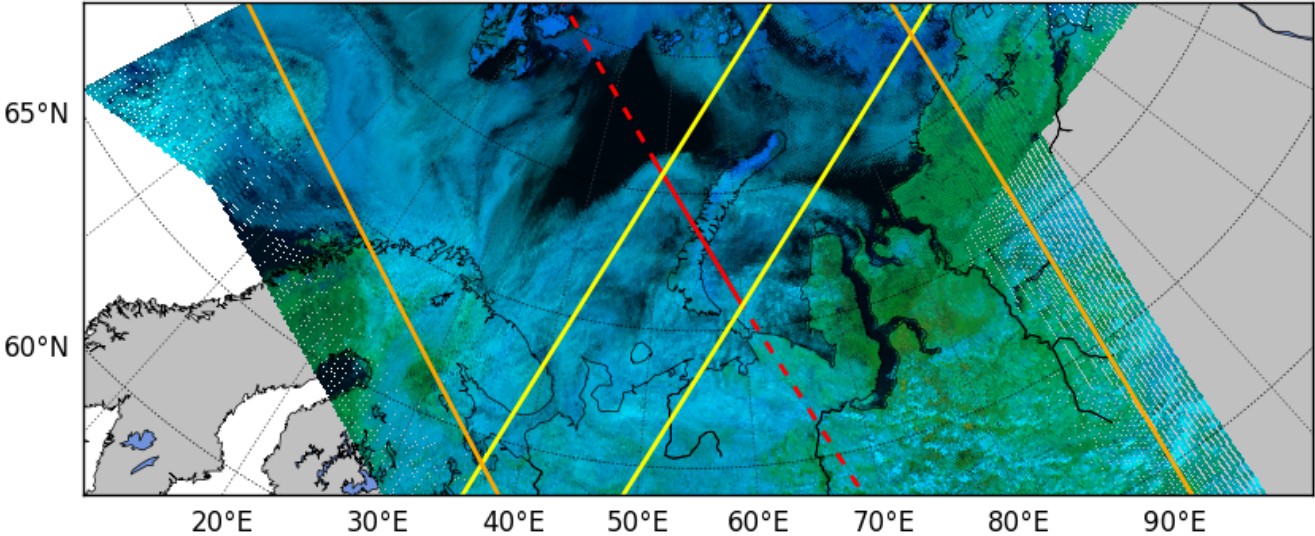

**Figure 12.** Study area SIB (Siberia). As Figure 08, but at 07/27/2008, 08:10 LST.

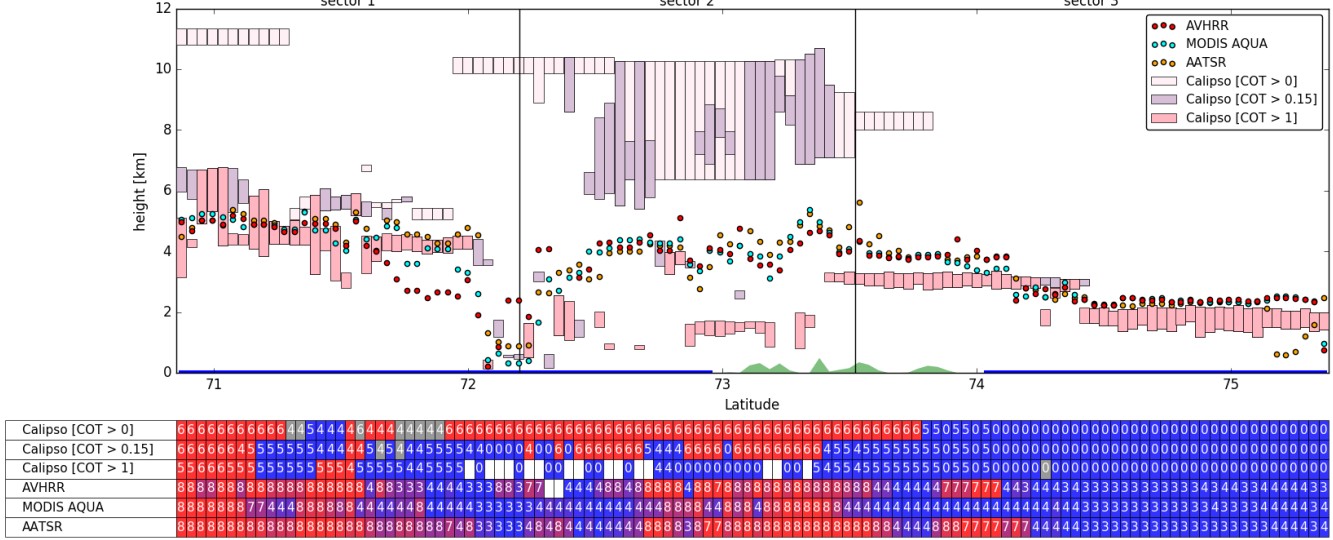

**Figure 13.** Study area SIB (Siberia). As Figure 09, but at 07/27/2008, 08:10 LST (n = 116).





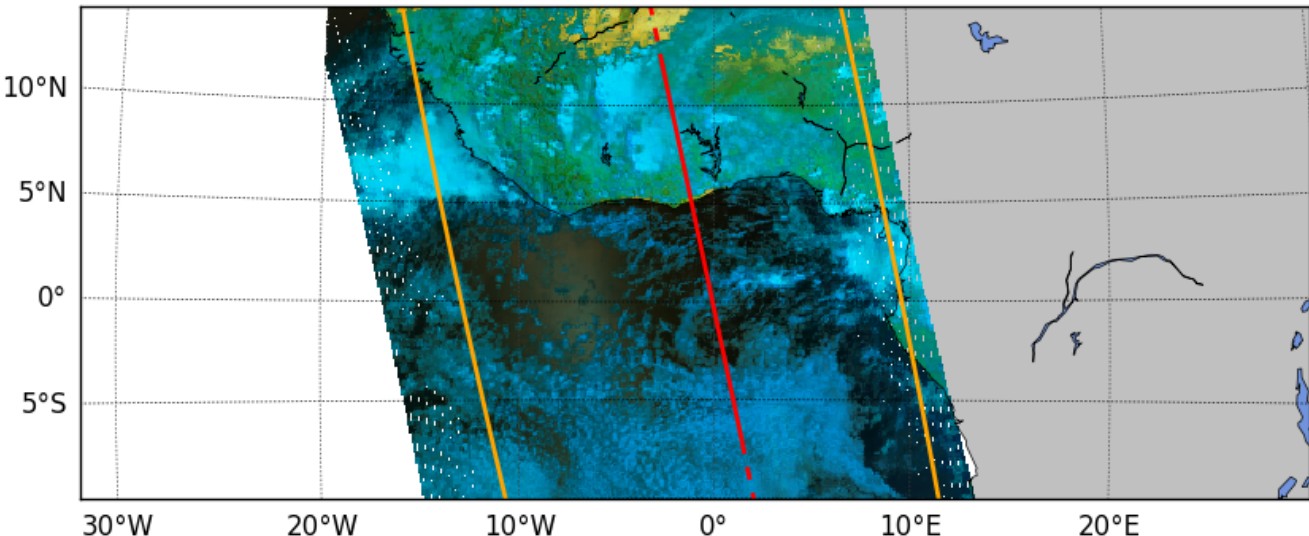

**Figure 14.** Study area AFR (Africa). As Figure 08, but at 10/24/2009, 13:45 LST.

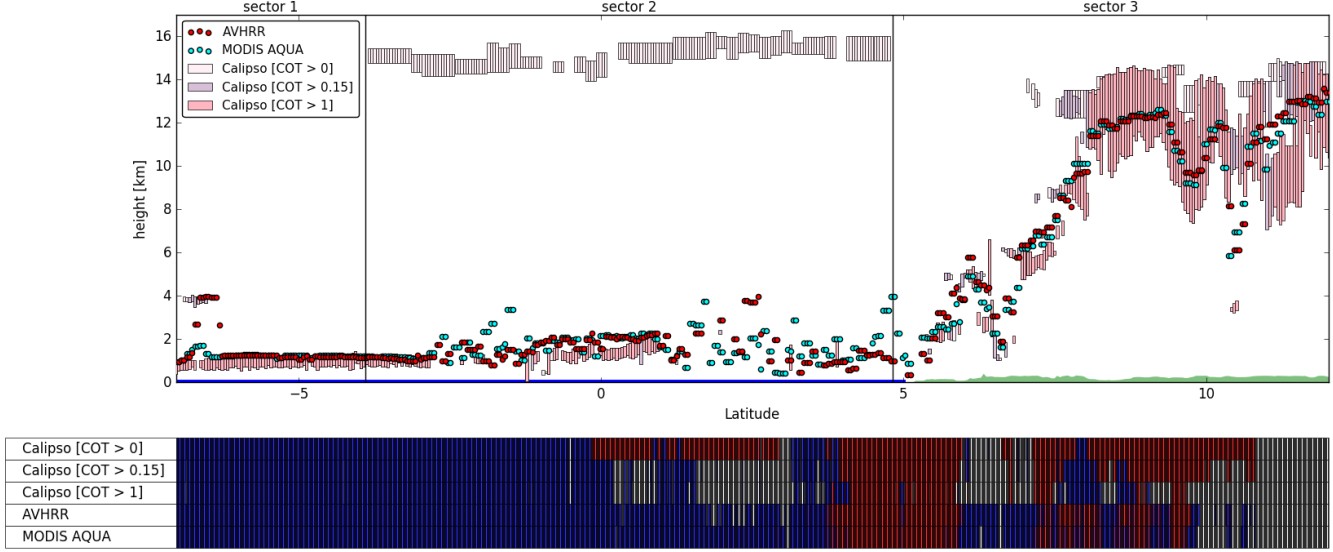

**Figure 15.** Study area AFR (Africa). As Figure 09, but at 10/24/2009, 13:45 LST. Due to space restrictions, no cloud type values are shown in table. n = 1181



**Table 1.** The CC4CL AVHRR-heritage dataset channel characteristics for AVHRR, AATSR, and MODIS. Instrument noise as applied within CC4CL is in reflectance for CC4CL channels 1-3, and in brightness temperature [K] for channels 4-6.

|        | CC4CL ID | sensor ID | channel width (µm) | noise |
|--------|----------|-----------|--------------------|-------|
| AVHRR  | 1        | 1         | 0.58 – 0.68        | 0.005 |
|        | 2        | 2         | 0.725 – 1.10       | 0.005 |
|        | 3        | 3a        | 1.58 – 1.64        | 0.005 |
|        | 4        | 3b        | 3.55 – 3.93        | 0.25  |
|        | 5        | 4         | 10.50 – 11.50      | 0.2   |
|        | 6        | 5         | 11.5 – 12.5        | 0.2   |
| MODIS  | 1        | 1         | 0.62 – 0.67        | 0.01  |
|        | 2        | 2         | 0.841 – 0.876      | 0.01  |
|        | 3        | 6         | 1.628 – 1.652      | 0.01  |
|        | 4        | 20        | 3.66 – 3.84        | 0.2   |
|        | 5        | 31        | 10.78 – 11.28      | 0.2   |
|        | 6        | 32        | 11.77 – 12.27      | 0.2   |
| AATSR  | 1        | 1         | 0.545 – 0.565      | 0.005 |
|        | 2        | 2         | 0.649 – 0.669      | 0.005 |
|        | 3        | 4         | 1.58 – 1.64        | 0.005 |
|        | 4        | 5         | 3.51 – 3.89        | 0.25  |
|        | 5        | 6         | 10.4 – 11.3        | 0.1   |
|        | 6        | 7         | 11.5 – 12.5        | 0.1   |



**Table 2.** Threshold values applied to ANNCOD data for cloud mask classification.

| day | night | twilight | land | sea | snow/ice | threshold |
|-----|-------|----------|------|-----|----------|-----------|
| x   |       |          | x    |     |          | 0.2       |
| x   |       |          | x    |     | x        | 0.35      |
| x   |       |          |      | x   |          | 0.1       |
| x   |       |          |      | x   | x        | 0.4       |
|     | x     |          | x    |     |          | 0.3       |
|     | x     |          | x    |     | x        | 0.35      |
|     | x     |          |      | x   |          | 0.2       |
|     | x     |          |      | x   | x        | 0.4       |
|     |       | x        | x    |     |          | 0.3       |
|     |       | x        | x    |     | x        | 0.4       |
|     |       | x        |      | x   |          | 0.35      |
|     |       | x        |      | x   | x        | 0.4       |



**Table 3.** Linear regression coefficients between collocated AVHRR and MODIS/AATSR channels.

| CC4CL channel ID | sensor | regression coefficients |
|---|---|---|
| 1 | MODIS | $0.8945 \times ch1 + 2.217$ |
|   | AATSR | $0.8542 \times ch1$ |
| 2 | MODIS | $0.8336 \times ch2 + 1.749$ |
|   | AATSR | $0.7787 \times ch2$ |
| 4 | MODIS | $0.9944 \times ch4 + 1.152$ |
|   | AATSR | $1.0626 \times ch4 - 15.777$ |
| 5 | MODIS | $0.9742 \times ch5 + 7.205$ |
|   | AATSR | $0.9793 \times ch5 + 5.366$ |
| 6 | MODIS | $0.9676 \times ch6 + 8.408$ |
|   | AATSR | $0.9838 \times ch6 + 4.255$ |



**Table 4.** Cloud type classification for CC4CL and CALIOP.

| ID | CC4CL | ID | CALIOP |
|----|-------|----|--------|
| 0 | clear | 0 | low transparent |
| 1 | switched to water | 1 | low opaque |
| 2 | fog | 2 | stratocumulus |
| 3 | water | 3 | low broken cumulus |
| 4 | supercooled | 4 | altocumulus |
| 5 | switched to ice | 5 | altostratus |
| 6 | opaque ice | 6 | cirrus |
| 7 | cirrus | 7 | deep convective |
| 8 | overlap | 8 | n/a |





**Table 5.** CC4CL primary and secondary output. NN = neural network, SV = state vector, PP = postprocessed, PV = Pavolonis et al. (2005) algorithm, OE = optimal estimation.

| variable name | abbrev. | unit | origin | description |
|---|---|---|---|---|
| | | | primary variables | |
| cloud mask | cldmask | 1 | NN | Binary cloud occurrence classification |
| cloud type | cldtype | 1 | PV | Categorical cloud type classification |
| cloud phase | phflag | 1 | PV | cloud phase classification |
| cloud top pressure | ctp | hPa | SV | OE retrieval estimate of cloud top pressure |
| cloud top pressure unc. | ctp_unc | hPa | SV | OE retrieval unc. of cloud top pressure |
| cloud effective radius | cer | μm | SV | OE retrieval estimate of cloud effective radius |
| cloud effective radius unc. | cer_unc | μm | SV | OE retrieval unc. of cloud effective radius |
| cloud optical thickness | cot | 1 | SV | OE retrieval estimate of cloud optical thickness |
| cloud optical thickness unc. | cot_unc | 1 | SV | OE retrieval unc. of cloud optical thickness |
| surface temperature | stemp | kelvin | SV | OE retrieval estimate of surface temperature |
| surface temperature unc. | stemp_unc | kelvin | SV | OE retrieval unc. of surface temperature |
| | | | secondary variables | |
| cloud mask unc. | cldmask_unc | 1 | PP | derived from NN output and threshold distance |
| cloud top height | cth | km | PP | derived from CTP and atmospheric profile |
| cloud top height unc. | cth_unc | km | PP | derived from retrieval unc. of CTP |
| cloud top temperature | ctt | kelvin | PP | derived from CTP and atmospheric profile |
| cloud top temperature unc. | ctt_unc | kelvin | PP | derived from retrieval unc. of CTP |
| cloud water path | cwp | g/m$^2$ | PP | derived from CER and COT (Han et al., 1994) |
| cloud water path unc. | cwp_unc | g/m$^2$ | PP | derived from retrieval unc. of CER and COT |
| cloud albedo at 0.06 μm | cla | 1 | PP | derived from CER and COT based on DISORT (Laszlo et al., 2016) |
| cloud albedo at 0.06 μm unc. | cla_unc | 1 | PP | derived from retrieval unc. of CER and COT |
| cloud albedo at 0.08 μm | cla | 1 | PP | derived from CER and COT based on DISORT (Laszlo et al., 2016) |
| cloud albedo at 0.08 μm unc. | cla_unc | 1 | PP | derived from retrieval unc. of CER and COT |
| cloud effective emissivity | cee | 1 | PP | derived from 10.8 and 12.0 μm data |



**Table 6.** Statistics of CTP, COT, and CER retrieval values for study area NA2 and AVHRR (first value in each cell), MODIS (second value), and AATSR (third value). Δ values are given for AVHRR minus MODIS (first value in each cell), AVHRR minus AATSR (second value), and MODIS minus AATSR (third value). *t-Test p-value > 0.1, indicating that differences in mean values are not significant.

|  | mean | median | stddev | skewness | kurtosis |
|---|---|---|---|---|---|
| CTP | 667.2, 665.0, 645.2 | 667.8, 668.1, 632.4 | 147.5, 142.7, 146.2 | -0.2, -0.2, 0.1 | -0.4, -0.4, -0.8 |
| Δ CTP | 2.2*, 21.9, 19.7 | 4.2, 22.3, 18.5 | 63.0, 138.7, 138.9 | -0.4, -0.3, -0.3 | 8.2, 1.0, 0.7 |
| COT | 12.3, 13.6, 13.4 | 7.2, 8.6, 8.8 | 19.8, 19.7, 17.6 | 6.6, 5.7, 5.3 | 60.5, 46.2, 40.8 |
| Δ COT | -1.3, -1.2, 0.2* | -0.6, -1.2, -0.5 | 16.5, 22.0, 21.3 | 0.7, 2.4, 1.8 | 59.6, 41.5, 33.1 |
| CER | 21.1, 19.2, 21.3 | 16.5, 14.4, 18.1 | 13.0, 12.1, 10.9 | 1.1, 1.4, 0.6 | 1.4, 1.2, -0.8 |
| Δ CER | 1.9, -0.2*, -2.1 | 0.5, -1.0, -1.9 | 7.0, 11.6, 11.3 | 0.8, 0.8, 0.5 | 7.9, 4.4, 2.3 |



**Table A1.** ERA-Interim variables used within CC4CL. Variables marked with * are available at 0.1°spatial resolution, all others default to 0.72°.

| variable name | abbrev. | ID | unit |
|---|---|---|---|
| **profile variables** | | | |
| Geopotential | Z | 129 | $m^2\ s^{-2}$ |
| Temperature | T | 130 | K |
| Specific humidity | Q | 133 | $kg\ kg^{-1}$ |
| Log. surface pressure | LNSP | 152 | Pa |
| Ozone mass mixing ratio | O3 | 203 | $kg\ kg^{-1}$ |
| **surface and single level variables** | | | |
| Sea-ice cover* | CI | 31 | (0-1) |
| Snow albedo | ASN | 32 | (0-1) |
| Sea surface temperature | SSTK | 34 | K |
| Total column water vapour | TCWV | 137 | $kg\ m^{-2}$ |
| Snow depth* | SD | 141 | m of water equivalent |
| 10 metre U wind component | U10M | 165 | $m\ s^{-1}$ |
| 10 metre U wind component | V10M | 166 | $m\ s^{-1}$ |
| 2 metre temperature | T2M | 167 | K |
| Land/sea mask | LSM | 172 | (0,1) |
| Skin temperature* | SKT | 235 | K |



**Table A2.** CC4CL L2 primary output variables. NN = neural network.

| variable name | abbrev. | unit |
|---|---|---|
| latitude | lat | degree |
| longitude | lon | degree |
| solar zenith | solzen | degree |
| satellite zenith | satzen | degree |
| relative azimuth | relaz | degree |
| cloud top pressure | ctp | hPa |
| cloud top height | cth | kilometer |
| cloud top temperature | ctt | kelvin |
| cloud liquid water path | cwp | g/m$^2$ |
| cloud effective radius | cer | μm |
| cloud optical thickness | cot | 1 |
| NN cloud optical thickness | cccot | 1 |
| cloud albedo | cla | 1 |
| cloud effective emissivity | cee | 1 |
| cloud fraction | cc_total | 1 |
| NN cloud mask | cldmask | (0,1) |
| cloud phase flag | phflag | 1 |
| Pavolonis cloud type | cldtype | 1 |
| retrieval convergence flag | conv | 1 |
| number of retrieval iterations | niter | 1 |
| a priori cost at solution | costja | 1 |
| measurement cost at solution | costjm | 1 |
| quality control flag | qcflag | 1 |
| land/sea flag | lsflag | (0,1) |
| snow/ice mask | siflag | (0,1) |
| illumination flag | ilflag | 1 |
| surface temperature | stemp | kelvin |





**Table A3.** CC4CL L2 secondary output variables. NN = neural network.

| variable name | abbrev. | unit |
| --- | --- | --- |
| cloud optical thickness a priori | cot_ap | 1 |
| cloud optical thickness first guess | cot_fg | 1 |
| cloud effective radius a priori | cer_ap | μm |
| cloud effective radius first guess | cer_fg | μm |
| cloud top pressure a priori | ctp_ap | hPa |
| cloud top pressure first guess | ctp_fg | hPa |
| surface temperature a priori | stemp_ap | kelvin |
| surface temperature first guess | stemp_fg | kelvin |
| albedo in channel no X | alb_ch_X | 1 |
| reflectance in channel no X | ref_ch_X | 1 |
| brightness temperature in channel no X | bt_ch_X | kelvin |
| firstguess reflectance in channel no X | fg_ref_ch_X | 1 |
| firstguess brightness temperature in channel no X | fg_bt_ch_X | kelvin |
| reflectance residual in channel no X | ref_res_ch_X | 1 |
| brightness temperature residual in channel no X | bt_res_ch_X | kelvin |
| degrees of freedom signal | deg_free | 1 |