# Peer review of "The Community Cloud retrieval for Climate (CC4CL). Part I: A framework applied to multiple satellite imaging sensors"

_Atmospheric Measurement Techniques, 2017_

## Referee Comment (RC1) · Anonymous Referee #2 · 15 Nov 2017

This manuscript provides an overview of the Community Cloud retrieval for CLimate (CC4CL) algorithm developed for the European Space Agency's (ESA) Climate Change Initiative (CCI) program. CC4CL is a cloud retrieval framework that can be applied to any number of passive satellite imagers having spectral information content sufficient to retrieve parameters such as cloud top (pressure, CTP), optical (cloud optical thickness, COT), and microphysical (cloud effective radius, CER) properties. For ESA's CCI, CC4CL is applied to AVHRR (NOAA-7 through Metop-A+NOAA-19), Terra and Aqua MODIS, ATSR-2, and AATSR, and uses a subset of solar and thermal IR spectral channels common to all sensors. Cloud identification, or masking, is performed using thresholds applied to COT derived from an artificial neural network

trained using NOAA-18 AVHRR and co-located CALIOP observations. Cloud typing (thermodynamic phase) is performed using the approach of Pavolonis et al. (2005), and cloud top, optical, and microphysical retrievals are performed simultaneously using the optimal estimation retrieval ORAC. Details on the required ancillary data, as well as the CC4CL pre-processor, are provided. The algorithm is applied to four case studies, selected for concurrent observations between the passive imagers and active CALIOP, for which retrievals of CTH (derived from CTP and ancillary atmospheric profiles) are evaluated with the co-located CALIOP cloud layer products; example CTP, COT, and CER retrieval swath images are shown for one case study. The authors show that CTH from each passive sensor is in general agreement, though there are some expected differences when compared with CALIOP.

The manuscript is clearly written, the figures provided are of excellent quality, and the sections are generally well organized. However, I have a number of comments, listed below, that should be addressed before accepting for publication. Thus the paper should be returned to the authors for revisions.

Comments

p 2, line 5: I would add cloud forward model assumptions to the list of secondary confounding factors.

p 2, lines 12-14: The CERES-MODIS products (e.g., Minnis et al., 2011a,b, IEEE TGRS) should also be included here.

p 2, lines 23-24: The MODIS C6 phase referred to here is the IR phase of Baum et al. (2012), which is in fact a quad-spectral algorithm (7.3, 8.5, 11, 12$\mu$m channels) using $\beta$ ratios (the authors' description is more appropriate for the C5 algorithm). This IR phase algorithm is run in conjunction with, and is informed by, the cloud top property retrieval algorithm. The authors should be aware, and I believe that they are given the reference to Marchant et al. (2016) later in the paper, that this IR algorithm does not determine phase for the C6 cloud optical properties retrieval; phase for the optical

retrieval is determined by the Marchant algorithm that uses the IR phase as one piece of information. Results from the IR and cloud optical properties phase algorithms are often at odds, specifically in cases where phase is more ambiguous.

p 2, line 24: Should probably specify that the additional spectral channels are at short-wave infrared (SWIR) wavelengths.

p 2, lines 29-30: Indeed, this is an inherent limitation of the spectral information content of passive IR channels!

p 2, lines 31-35: I assume from the references given that cloud cover refers to cloud fraction or related metrics, and not to geophysical retrievals.

p 3, line 5: Is the cloud phase bias positive or negative?

p 3, lines 6-7: See my p 2 comment above regarding MODIS phase algorithms; this statement again refers only to the IR phase.

p 3, lines 10-11: What is the difference between consistency and continuity? I can surmise that it is consistency in approach versus continuity of results, but it is not clear to the general reader.

p 3, lines 34-35: It's not initially clear why independent retrievals of COT/CER and macrophysical products are inherently radiatively inconsistent. I would guess that it depends on the approach, i.e., how (or if) one set of retrievals informs the retrieval of the other. Can the authors better explain?

p 4, line 1: Retrieval uncertainty estimates that propagate errors is not a novel feature of CC4CL. See, for instance, the MODIS C6 cloud optical properties (Platnick et al., 2017), which provide pixel-level retrieval uncertainties calculated in a manner that is mathematically consistent with that of optimal estimation (although the uncertainties are not part of the solution process).

p 4, line 6: Following on my comment above, neither the optimal estimation approach

nor the uncertainty quantification are novel features of CC4CL. As the authors themselves state on p 2, PATMOS-x uses optimal estimation theory, and the MODIS C6 (and C5) cloud optical properties provide rigorous pixel-level uncertainties.

p 4, lines 5-13: Regarding statements about consistency of the long-term, multi-platform time series, and the potential of the framework for climate studies, I don't think the authors make a convincing case for either in the text that follows. Four case studies hardly constitute a "comprehensive and detailed analysis of retrieval results," and certainly do not provide enough evidence of the potential for climate studies. Such statements require detailed analyses of long-term and large-scale inter-sensor statistical comparisons, which it appears are actually presented in a companion paper in a different journal (Stengel et al., 2017). It's thus not clear to me why the present paper was not instead a part of the Stengel paper, or vice versa. Given that the primary contributions are a brief discussion of the ancillary and data sources and a rather limited CTH analysis, I'm not convinced that this paper can or should stand on its own.

p 4, line 15: Consider using Level-1 instead of L1, which for some readers implies a Lagrange point 1 orbit.

p 4, lines 21-25: Yes, replacing any AVHRR once its successor becomes available will lessen the impacts of orbital drift (and thus sampling times), but drift impacts are likely still to exist. Are these accounted for in CC4CL, specifically when constructing long-term multi-sensor time series?

p 4, line 29: Regarding filtering channel 3b data, is this to include or exclude that channel?

p 5, lines 8-10: It should be NASA Goddard Space Flight Center.

p 5, lines 21-23: "Self-calibrating" is I think a little misleading. MODIS, for instance, has a similar design (onboard black bodies and solar diffuser), yet requires a continual effort to monitor instrument stability and identify/correct calibration drifts, typically using
fixed ground targets among others.

p 7, lines 3-4: Has the "gap filling" of the MCD43C1 data been validated? Is the approach similar to what is used in the MCD43B3 gap-filled product (Schaaf et al., 2011, "Aqua and Terra MODIS albedo and reflectance anisotropy products," in Land Remote Sensing and Global Environmental Change: NASA's Earth Observing System and the Science of ASTER and MODIS)?

p 6-7, Sections 2.2.3-2.2.4: Have the authors verified that there are not any trends in the land surface BRDF and emissivity time series during the MODIS era? If there are, wouldn't the use of the climatology derived from all MODIS data introduce a discontinuity in the surface time series?

p 7, lines 6-7: I disagree that the surface is a minor component of the observed signal, specifically for optically thinner clouds. Thus not accounting for the spectral response functions can introduce biases, particularly in spectral regions such as the near-IR (e.g., AVHRR channel 2, MODIS channel 2) where reflectance by vegetation can change rapidly.

p 7, line 16: Resampled or aggregated?

p 7, line 16-17: I would agree that differences in sensor spatial resolution are reduced when averaging radiances/reflectances. However, this is likely not the case when averaging L2 geophysical parameters, as is done here, since the retrievals can have significantly different PDFs within a grid box due to pixel size differences alone.

p 9, line 5: How much data was used to train the ANN? Was an observation time difference filter applied to the NOAA-18/CALIOP co-location?

p 9, lines 19-21: If I understand correctly, the reflectances/radiances were adjusted to account for spectral response differences? Were the co-located observations filtered for cases in which both satellites viewed the scene at the same sun-view angle geometry? Such angle matching is important when comparing solar channels where

reflectance is strongly angularly dependent.

p 10, lines 21-22: My understanding is that the uncertainty obtained from the optimal estimation framework can be thought of as the sensitivity of the solution space at the point of the solution to the measurement uncertainty (which includes instrument, ancillary, etc., uncertainties).

p 10, line 25: This statement differs from the statement at the end of Section 3.2 (phase is determined first to reduce computation time resulting from retrieving assuming both phases).

p 11, Section 4.1, Figure 3-5. The observation date/times should be stated here. I see they are listed in Section 4.3, but it is better to include them at first reference. Also, a thermodynamic phase image would be useful.

p 11, lines 13-14: I'm guessing the peaks at 12 and 35 $\mu$m likely correspond to liquid and ice phase clouds, respectively.

p 11, line 17: The statement on cloud displacement here contradicts the statement in line 11.

p 11, Section 4.1: What about relative radiometric calibration between the different sensors? Even minor differences of a couple percent could case large retrieval differences, particularly for COT.

p 11, line 22: If median absolute CER uncertainty is $2\mu$m, how does this correspond to a median relative uncertainty of 2% (line 24).

Figure 10: What wavelengths are used for this RGB?

p 12, Section 4.3: Hard to call this "validation" without using a much larger dataset (e.g., months, seasons, years) for statistical analyses.

p 12, line 21: What assumptions are made other than adiabaticity (e.g., extinction profile, etc.)? Also, what does adiabaticity mean for an ice phase cloud?

p 12, Case Study NA1: Need to include the Figure number in the text.

p 13, lines 25-26: Which existing algorithms were compared to these results?

p 14, lines 10-11: Why not show the extensive validation here?

p 15, lines 6-11: For the optimal estimation retrieval, are the spectral response differences handled similar to the ANN cloud mask (i.e., adjustment factors), or are they explicitly included in the forward model? What about relative radiometric calibration, could that be playing a role in the large MODIS-AATSR retrieval differences?

p 15, line 18: Here calibration deficiencies are acknowledged. Relative calibration should be explored as a cause of the retrieval differences.

p 15, lines 29-30: Can the authors provide references for these user applications?

p 16, line 29: "radiatively effective rather than physical cloud top"

p 17, line 9: The MODIS C6 phase referred to here is that of the cloud optical properties algorithm, not the IR phase referred to earlier in the paper.

p 18, lines 10-12: Perhaps this is worded poorly? I would imagine that real, complex vertical cloud structure is in fact a large source of retrieval errors, but the analytical approach to retrieval uncertainty used here (and in other retrievals) cannot account for this.

---

## Referee Comment (RC2) · Anonymous Referee #1 · 31 Dec 2017

This is a mature manuscript that describes a new framework for retrieving cloud properties from passive imaging sensors, which are partially validated by / trained with active techniques (CALIPSO). The main thrust of the paper is to ensure that one single technique be applicable to a range of sensors (mainly AVHRR/MODIS). The stated goal is to maximize the length of the time series available from the collection of the various sensors in low earth orbit (the approach is only applied to polar orbiters, not to geostationary satellites). For this reason, only "heritage" channels are used, which has the clear disadvantage that newer developments (such as the 1.38 micron channel for the detection of thin ice clouds or the 2.1/2.25 micron channel for better phase discrimination in MODIS/VIIRS) cannot be taken advantage of. The authors acknowledge that

they made a conscious decision to do so, but this also begs the question about the distinct benefit of this particular manuscript over already existing Climate Data Records such as that based on PATMOS-x. The authors do discuss prior efforts extensively, but don't really answer the question why we need yet another multi-satellite retrieval framework, except to say that ESA solicited the creation of Essential Climate Variables within their Climate Change Initiative. There are a few novel aspects in this approach, which are not always sufficiently explained (details see below, sequential comments). Beyond those, however, the manuscript does not conceptually take us beyond ISCCP and more modern climatologies. A truly novel retrieval would move beyond a single-pixel approach and consider the context and geographic region for increasing the information content in cloud retrievals. It would also not apply traditional optimal estimation without careful consideration of non-linearities (this has be done with Bayesian approaches such as Markov Chain Monte Carlo sampling - see manuscripts by Posselt). The authors did replace the cloud mask with a Neural Network approach tied to CALIPSO, and this seems meritorious because it seeks to objectivize passive imagery based retrievals by using active techniques as independent data source. Other than this important innovation, it remains unclear whether CC4CL truly is an improvement over existing techniques (such as PATMOS-X), or simply re-creates such efforts with slight modifications. Despite this concern, the collocation and cross-comparison of multiple instruments is convincing, and the fact that the code is presented as an open-source development makes the work very compelling. The next manuscript version should include information where the code and documentation can be downloaded.

The manuscript is rather heterogeneous in terms of the language quality. Details are given along with the sequential comments below. The most important (major) comment is that about the use and interpretation of ANN for the cloud mask. Generally, more specificity will be required in multiple aspects. Most of them will be possible to implement through minor revisions, but it would be good to have the manuscript go through another brief review after implementing them.

Sequential comments (minor and major comments are mixed):

p1: "Climate data record" is not discussed. It is also not clear until page 2 that the manuscript does indeed seek to develop a Essential Climate Variable (although it is unclear without reading Hollmann 2013 which of the 13 ECVs CC4CL will contribute to). A discussion of "essential climate variable", "CDR", and how this work fits in should be better discussed. It should also be discussed how it distinguishes itself from existing efforts in this regard (e.g., https://www.ncdc.noaa.gov/news/new-cloud-properties-climate-data-record).

p1l38: "shielding" is not a radiative transfer term - what is it in this context? Isn't it the same as "forcing"? Why are both terms used?

p1l38: "forcing": There is a difference between "radiative forcing" and "radiative effect" - which are the authors referring to? Probably the latter.

p1l49: This sounds like the variables "propagate uncertainties" into the derived cloud properties, which would be incorrect.

p2,l14: While "auxiliary" instead of "ancillary" data have become almost interchangeable, the latter is more correct; "auxiliary" has the connotation of only being a replacement in case the "primary data" is not available (compare: auxiliary power, not ancillary). For satellite retrievals, ancillary expresses more accurately that data from other sources are ingested within the operational algorithm.

p2,l25: "...not guaranteed to be radiatively consistent with..." It is unclear what that means (although the reviewer agrees with the statement). Please provide references. Also, does CC4CL perform "better" in terms of radiative consistency?

p2,l39: "sees" into the cloud: A retrieval is not animate. Replace colloquial "see" with more appropriate wording.

p2,l50: CONUS = contiguous US (conterminous is synonymous, but used much less frequently, also not by Sun et al., 2015).

p2,l55-l57: This is an important statement: Cloud cover is not a good observable for trend detection because it depends on its definition (optical thickness threshold and/or reflectance threshold, sensor resolution) and instrument performance or calibration drifts. Even the CALIPSO-derived cloud information depends on which resolution is considered (because of sensitivity and SNR). A better observable would be the optical thickness itself (or better still, the cloud radiative effect). Have the authors considered a different primary variable that is more amenable to trend detection than cloud cover? In fact, their approach of retrieving "pseudo CALIPSO optical thickness" seems to be going exactly in this direction - and in the reviewer's opinion, this would be the right way to proceed. But why then go a step backwards and convert ANNCOD into a binary cloud mask? Why isn't the retrieved ANNCOD not reported directly (in addition to the binary cloud mask outcome)?

Related to the above [and also to material on p6]: Since CC4CL does keep cloud cover as primary variable, it should be explained whether the thresholds (table 2) vary (for example, with the specific sensor or orbit), or whether they are fixed once and for all, now that they have been optimized via the ANN technique. More importantly, do the weights as established during the ANN learning process vary? Are they a function of orbit, instrument, illumination, surface, topography...? Or else, are all of these dependencies incorporated in one single ANN? If so, how are commonly known problems with ANN (such as overfitting) avoided here? Using this cloud masking and thresholding technique, what is the (minimum) cutoff optical thickness, below which cloud are no longer detected? How do optical thickness detection thesholds vary with surface type and sun-sensor geometry?

Related to the above [and also to material on p6]: The three elements of the ANN need to be described better. How well is the pseudo-CALIOP optical depth itself estimated with the ANN? Figure 2 illustrates the performance of the cloud mask after ANNCOD has be converted into a binary cloud mask. Since the ANN predicts ANNCOD and not the cloud mask itself, it should be the performance of the ANN with respect to

ANNCOD that should be demonstrated here. In this context again: How is overfitting avoided? How can the non-linearities of radiative transfer be emulated with a single hidden layer? What is the result for ANNCOD for the training data set as opposed to the test data set? How is the correction for viewing angle done? How many inputs does the input layer have; what are they? What is the activation function? Are there bias perceptrons? What motivates the use of one single hidden layer, and why are there 50 neurons in it? Is the network re-trained for every new satellite data set, or are the weights fixed? How exactly were the threshold values from table 2 determined that are applied to ANNCOD to translate into cloud mask?

Finally, what is the quality of the thermodynamic phase retrieval, optical thickness and effective radius, depending on how close ANNCOD is to the cloud detection threshold? Essentially, the paper claims that a cloud retrieval is attempted if the optical thickness exceeds 0.4 over snow/ice during day light conditions. This would be a remarkable improvement over existing retrievals. MODIS usually does not detect clouds over snow-covered areas in the Arctic unless they have an optical thickness significantly larger than 0.4 (around 7). CC4CL would be an improvement of an order of magnitude, and the question is whether the cloud retrievals would be of practical use, especially when applying them to AVHRR instead of MODIS. The reviewer strongly believes that the only way to achieve detection thresholds on the order of 0.4 in optical thickness in snow/ice covered regions in the Arctic, one would need to use convolutional layers (i.e., use multi-pixel retrieval approaches).

p2,l78-80: "Consistency can be traded for continuity" needs clarification. Perhaps this can be done while elaborating on CDR (see comment above). This discussion will contribute to a better motivation of this study.

p2,l90: "MODIS provides": is a partial repetition of material in the left column of the same page.

p3,l18: "on other" > "over other"

p3,l20: Which "macrophysical" product do the authors have in mind here? What exactly does "radiative inconsistent" mean (supposedly, macroscopical products are inconsistent with microphysical products, but this is different from "radiatively inconsistent"; the reader is currently left to guess here). How exactly does the CC4CL approach ensure radiative consistency amongst all input satellite radiances (and all output products)? Indeed, other approaches have a cloud mask that may be independently derived from the microphysics products. Simply stating that CC4CL is "different" in this regard does not support the statement that it is more "consistent". More details are needed to add specificity.

p3,l46: Quantify "very realistic", or just use "realistic"

p4,l35: Auxiliary > Ancillary

p4,l38: Neural Network not yet defined at this point. May need the NN section prior to this statement.

p4,l73/l75: "optimal estimation", "cloud typing scheme". None of these have been described at this point in the manuscript. Sequence needs to be re-shuffled.

p5,l1: "were" > "are"

p5,l1: Reference and/or data source (link) needed for CALIPSO product

p5,l68-l72: multiple acronyms need to be introduced prior to first use.

p5,l79: The outcomes of the study should be at least summarized here. Also, the use of "round robin" may not be ideal for an international readership as it is a cultural reference (British/American) that may not be commonly known. Consider paraphrasing the technique instead.

p5,l98: Do these channel numbers refer to the CC4CL IDs from table 1?

p6: Cloud detection: See multiple comments above (following p2,l55-l57 comment) Also: Are there any convolutional layers included in the approach? This would have

allowed capitalizing on the context of a pixel.

p7,table 3: How was the regression done - based on radiance or irradiance, based on counts? Based on brightness temperature (for IR channels)? The offsets seem rather large; what is the explanation for significant offsets?

p7,l49: VIIRS algorithm is used: What is the purpose of this statement? If it is kept, this needs to be elaborated (what does the VIIRS algorithm do differently). Also, there are various other algorithms that are improved over the heritage algorithms, which would probably all need to be mentioned here (or at least a subset thereof).

Figure 2: This is just one example where labels are too small, and are too pixelated. Generally improve the figure quality and enlarge labels. About the content: It is rather hard to interpret this figure. The x-axis is "normalized". Does that mean that the difference of the ANNCOD-retrieved value and the threshold from table 2 is divided by the threshold value itself? Does "x=0" mean that the retrieved optical thickness equals the threshold per table 2? Does the "CLEAR" label refer to CALIPSO? For x=-0.2, we find an uncertainty of 40%. Does that mean that CC4CL misclassifies clear pixels as "cloudy" in 40% of cases?

p8,l70: Why are largest uncertainties found for opaque clouds? Also, figure 10 does not show quantitative evidence for this statement - colors are harder to interpret than numbers on a graph. Can this somewhat counterintuitive statement be supported by a more succinct graph?

p9,l3-5: :Validation is show for . . . rather than: Unclear. What is the difference between CTH and "its" retrieved value?

p9,l13: "TOA radiation is the *sum total* of emission and scattering throughout the atmospheric column" - please formulate this more accurately: What is a "sum total" of two processes? Also, the next paragraph more or less paraphrases Platnick's vertical weighting function paper where this is formulated more accurately, and where the

concept of a weighting function is well explained. Please cite that paper and use similar terminology here. As for multi-layer clouds, there is a fairly new paper by Wind, Platnick et al. (http://journals.ametsoc.org/doi/abs/10.1175/2010JAMC2364.1), but it is probably not applicable to this paper here because of the channel selection.

p10,l1: How is the CTH adjustment done if the cloud base is not known? Where does cloud base (or cloud geometrical thickness) information come from?

p10,l9: Does this statement about sectors refer to figure 9? Please match figures and text, otherwise figures become "orphans" that are not tied to the manuscript.

p10,l14: Please define what is meant by "surface" in this case.

p10,l16: insert "a" before "single-layer"

Figure 7: please enlarge labels, as well as histograms; it is hard to compare the retrievals quantitatively otherwise. Also: It would really help if histograms were shown separately for snow-covered areas as opposed to dark surfaces. It is expected that retrieval quality would differ significantly depending on the surface conditions.

p12,l41: "performance of existing algorithms" What are the "existing algorithms" that CC4CL? Has the manuscript shown that these existing algorithms perform less well than CC4CL.

p12,l88: "AVHRR" > "for AVHRR"

p12,l89: Should "continually" be replaced with "consistently"? Unclear what this statement means. If it were "consistently" it would be more clear, but the word order should be fixed: "The CC4CL phase identification does not agree with any of the three CALIOP cloud flags consistently, which is reasonable given ..."

p13,l19/20: "...insensitive to the specific instrument evaluated, such that the merged data set is sensible". What does this statement mean? The paper does not actually present a *merged* data set, or was that the actual intent of the paper? It does evaluate

collocated overpasses from different satellites, but these are not merged in the sense of a CDR. Please remove the statement about "merging" data sets unless this was the actual intent of the paper (in which case it would need to be modified considerably).

p13,l31: "disagree nonetheless": They disagree despite their channels are fairly close? Can this be re-phrased? The whole paragraph is a bit roundabout. There's a 30-40% difference in reflectance, but "their" retrieval values are "much more similar"? Please make this statement more precise. "The difference to AVHRR and MODIS is largest for CER" - does this statement refer to AATSR again?

p13,l39: The t-test needs to be explained in much more detail. What is H0, what is mu1, what is mu2? Are we talking about the covariance between two data sets, which is assessed using the t-test approach? If so, are the data from the two different data sets (supposedly this is what "mu1" and "mu2" refer to) re-gridded to one common grid before comparing them? The premise of this statement deserves at least one paragraph, if not half a page.

p13,l45: "spatiotemporally collocated sensors": The sensors are not collocated - is that the point of the statement? Or is this an explanation why the t-test "fails? What does "non-significant" t-test mean? Could the strictness of the comparison be relaxed by gridding the retrievals to a coarser common grid before making the inter-comparison?

p13,l57: "depending on the user's application" - this needs to be clarified. For which applications can they be used interchangeably? Could a combined AVHRR and MODIS cloud data record constitute a CDR (would it meet the requirements)? As stated above, the manuscript does not actually "merge" data sets in this way, but more specificity would be helpful here.

p13,l77: "we see that COT uncertainty scales with COT itself": this is not shown in the manuscript. If it is, please refer to a figure or section.

p13,l79-l88: Consider re-writing this section; simplify and use literature references;

[Figure]

most of these observations have been documented before (large COT uncertainty as reflectance approaches asymptotic value; large uncertainties for bright surfaces).

p15, l11: "otherwise are" > "otherwise they are"

p15,l15: "may it stem" does not work in English; consider "whether it stems from. . . or"

p15,figure 11: The table below the cross section is too small. Also, what happened at lat=61? Why do the active imagers pick up a cloud where CALIPSO does not?

p16,l8: consider "a conscious decision was made to [deliberately] trade. . ."

p16,l19: "on a first view" > "at first glance"

p18,l59: "synergic" > "synergistic"

p18,l95: "accurate and precise": These two were not discussed separately. Where was this done? If not, please clarify this statement.

---

## Referee Comment (RC3) · Anonymous Referee #3 · 8 Jan 2018

Review of "The Community Cloud retrieval for Climate (CC4CL). Part I: A framework applied to multiple satellite imaging sensors", by Sus et al.

The manuscript introduces a valuable approach to establish a common passive cloud retrieval applicable to a series of standard polar orbiters in order to create data sets usable for climatological studies. This would be an important step for the community and the usability of satellite products outside the satellite community. I also understand and acknowledge the need to base such an approach on well established methods instead of more experimental approaches as suggested by one of the other referees. The general presentation is of good/excellent quality. My two co-referees have elabo-

rated on a number of specific technical and scientific details already. I want to focus on a more general weakness.

What exactly is the focus of this manuscript? If I missed important, clear, early statements in the existing text, I apologize. If not, the reader needs this guideline.

In many places important details can not be given and are not explained owed to the shear extent of this project. In most cases the reader is then correctly referred to other publications where the methods of CC4CL are introduced. This way the purpose of the manuscript at hand becomes more and more unclear while reading through it. First impression is that the general method will be explained. But then the core retrieval techniques are explained elsewhere (McGarragh). Then a technical explanation of the ANN cloud mask is started, but it stays too short to be fully comprehensible. After the introduction of example cases Fig 3-5 and cross sections Fig 9-16, I expected an in-depth discussion of reason for differences and a quantitative validation (section titles containing "validation") or cross-comparison of all products, but the discussion stays very general and mostly describes differences. Proper validation is again shown elsewhere (Stengel).

The limited original content of this manuscript (correct me, if I'm wrong) is not reflected by the title and manuscript length (e.g. 8 figures 8-15 with very comparable content and not too surprising differences between active and passive sensor, but no quantitative validation). The authors should clarify the purpose of this manuscript and shorten parts published elsewhere even stronger. I suggest to consider these general points and a revision of the manuscript.

Specific major issues:

p3, line 27: "Moreover, the resulting time series are carefully validated ... (ISCCP, PATMOS-x, CM SAF, and MODIS Collection 6), reanalysis and model data (ERA-Interim and EC-Earth), ground-truth synoptic observations, and CALIOP lidar data." My understanding was that I would see that in this manuscript: You will only show

CALIOP comparisons, will you? Could you please clarify.

p8, section 4.3: I think you cannot call this chapter "validation". There is no systematic validation, only a few selected case studies, which mainly show the problems and no systematic quantitative validation. Four case studies of time height cross sections are shown only to present that lidar cth does not have much to do with passive cth? I also expected CER and COT validation somewhere.

p13, line 15: You mean, proper quantitative validation is shown in another paper ... Stengel et al 2017 ESSD? The retrieval method was shown in two other papers as well ... McGarragh 2017 at JAS and AMT. Remind me about the reason for this manuscript?

Minor issues:

p 2, line 33: AVHRR was not introduced before, was it?

p 2, line 53: What is r?

p 2, line 65: How can CTP and CTH be underestimated at the same time? Can you please comment?

p 2, line 66: What is a "cloud phase bias ... of 9%"? Cloud phase? Liquid and ice? Or cloud cover?

p 2, line 68: Low or hight bias?

p 2, line 102: It would be nice to say at this early stage what the purpose of this particular manuscript is in ESA Cloud_cci? And what other parallel publications contribute? Later on, the reader gets the impression that everything relevant is introduced elsewhere.

p 6, line 27: If this is the only description of ANNCOD available, you might at least want to cite Kox et al . 2014 (AMT, 7, doi:10.5194/amt-7-3233-2014) who introduced the idea and described in much more detail.

p6, line 51ff: This is all a slightly vague description, if it isn't detailed somewhere else. Why do you need ... after viewing angle dependency correction ... a whole set of thresholds? ANNCOD already gives an answer on the question cloud or no-cloud, doesn't it?

p7, Figure 2: y-axis. It is PEC not 1-PEC shown, isn't it? Does the graph show that, at your threshold you are only correct by about 50%?? Please discuss.

p10, line 10: "consistent". You could also say its all over the place, with different physical reasons in any single column. This is not a validation. You even tried to correct cth for cc4cl and still have big problems.

p11, Figure 7: Please make the labels consistent with the rest of the manuscript: n18->avhrr, myd->modis ...

---

## Author Comment (AC1) · 5 Mar 2018

RC: referee comment

AR: author response

AC: author's changes in manuscript
* * *
RC: p1: "Climate data record" is not discussed. It is also not clear until page 2 that the manuscript does indeed seek to develop a Essential Climate Variable (although it is unclear without reading Hollmann 2013 which of the 13 ECVs CC4CL will contribute to). A discussion of "essential climate variable", "CDR", and how this work fits in should be better discussed. It should also be discussed how it distinguishes itself from existing efforts in this regard (e.g., https://www.ncdc.noaa.gov/news/new-cloud-propertiesclimate-data-record).

AR: We will move p2 lines 97-106 to the very beginning of the introduction, and add some text to elaborate how our study fits into ESA CCI and the production of ECVs. We will now also mention CDR, but we do not elaborate on it, as it is not an essential part of this paper. We thus refer to Stengel et al., 2017, where this concept and resulting data have been published.

AC: "The European Space Agency has established the ESA Climate Change Initiative program (ESA CCI, 2015; Hollmann et al., 2013) in order to advance knowledge of the climate system through the generation of satellite based data records utilizing European and non-European assets. The CCI project's primary focus is the production of thirteen Essential Climate Variables (ECVs) covering ocean, atmosphere, and land geophysical variables. With these data records, CCI is aiming to fulfil highest climate requirements from the Global Climate Observing System (GCOS). This study presented here is part of the ESA CCI for clouds (ESA Cloud_cci), which has the objective to develop a state-of-the-art open-source community cloud retrieval algorithm which shall be capable of processing passive satellite imager data covering several decades. Both in part I and part II of this paper, we present the processing framework as developed within ESA Cloud_cci (CC4CL, part I), the detailed mechanisms of the optimal estimation retrieval (part II), and provide an initial assessment of the strengths and weaknesses of derived cloud parameters (part I). With CC4CL, several decades of passive imaging satellite data have been processed and are made available to the user. The resulting climate data records (CDR) are presented in Stengel et al., 2017."

AR: We will amend p2, line 106ff.

AC: p2, line 106ff: "In order to produce the cloud CDR presented here, we used satellite data from MODIS…"

RC: p1, l38: "shielding" is not a radiative transfer term - what is it in this context? Isn't it the same as "forcing"? Why are both terms used?

AR: We agree that shielding is superfluous, will remove.

AC: "Clouds considerably influence the global energy budget through direct radiative effects."

RC: p1, l38: "forcing": There is a difference between "radiative forcing" and "radiative effect"- which are the authors referring to? Probably the latter.

AR: Agreed, we are referring to radiative effect.

AC: See comment above.

RC: p1, l49: This sounds like the variables "propagate uncertainties" into the derived cloud properties, which would be incorrect.
AR: Will clarify.
AC: "Several secondary variables (state of surface and atmosphere, viewing geometry, sensor calibration and spectral response uncertainties) further complicate cloud retrievals, and insufficient knowledge on their state propagates uncertainties into the derived cloud properties."

RC: p2,l14: While "auxiliary" instead of "ancillary" data have become almost interchangeable, the latter is more correct; "auxiliary" has the connotation of only being a replacement in case the "primary data" is not available (compare: auxiliary power, not ancillary). For satellite retrievals, ancillary expresses more accurately that data from other sources are ingested within the operational algorithm.
AR: Agreed, will replace auxiliary with ancillary throughout the text.

RC: p2,l25: ". . .not guaranteed to be radiatively consistent with. . ." It is unclear what that means (although the reviewer agrees with the statement). Please provide references. Also, does CC4CL perform "better" in terms of radiative consistency?
AR: The effect of COT/CER/CTH on the top-of-atmosphere (TOA) radiances differs between the different sensing bands as a function of atmospheric state. For example if you used just the 11 or 12 micron measurement to estimate CTH then you must assume something about the COT (usually that it is thick) and something about the CER (typically a climatological value). If the COT assumption is incorrect (e.g. cloud is not thick) so that more upward radiance is transmitted through the cloud than expected then the cloud top appears too warm and is located (incorrectly) lower in the atmosphere. On the other hand using an all channel fit will identify the cloud as optically thin (from the visible and near visible reflectance measurements) and will avoid this error.

We note that retrieving a specific cloud property from a specific channel is radiatively inconsistent (as example above) but it is theoretically possible to do a sequential optimal estimation retrieval. In this case one iterates through the channels improving the estimates of CTH/CER/COT with each step. The final result should be the same as an all channel optimal retrieval. This method is not adopted for our problem as it would be computationally less efficient.

See also the introduction in part II of this paper for a detailed definition a radiative consistency. We will not elaborate much on the issue here, as that already happened in part II.
AC: "However, the derived microphysical variables are not guaranteed to be radiatively consistent with independently derived cloud parameters, as most of the retrieval methods are separated into solar and thermal methods even though measurements in these spectral regions are not independent of parameters retrieved in the other."

RC: p2,l39: "sees" into the cloud: A retrieval is not animate. Replace colloquial "see" with more appropriate wording.
AR: Will rephrase.
AC: "and beyond a penetration depth into the cloud corresponding to > 1 cumulative optical depth."

RC: p2,l50: CONUS = contiguous US (conterminous is synonymous, but used much less frequently, also not by Sun et al., 2015).
AR: Will rephrase.
AC: "and contiguous United States"

RC: p2,l55-l57: This is an important statement: Cloud cover is not a good observable for trend detection because it depends on its definition (optical thickness threshold and/or reflectance threshold, sensor resolution) and instrument performance or calibration drifts. Even the CALIPSO-derived cloud information depends on which resolution is considered (because of sensitivity and SNR). A better

observable would be the optical thickness itself (or better still, the cloud radiative effect). Have the authors considered a different primary variable that is more amenable to trend detection than cloud cover? In fact, their approach of retrieving "pseudo CALIPSO optical thickness" seems to be going exactly in this direction - and in the reviewer's opinion, this would be the right way to proceed. But why then go a step backwards and convert ANNCOD into a binary cloud mask? Why isn't the retrieved ANNCOD not reported directly (in addition to the binary cloud mask outcome)?

AR: Here, we are referring to cloud cover as one of several other variables that were analysed for other retrieval frameworks in separate studies. In this study, we are only presenting a retrieval framework and an initial assessment of its data quality. We do not present trend data, but rather refer to other studies that assessed quality of other CDRs, including cloud cover but also CTH and CTP. The ANNCOD is a temporary retrieval product, from which we derive cloud cover. Cloud cover information is used to avoid processing cloud-free pixels and thus to reduce processing time. ANNCOD data are contained in ESA Cloud_cci L3U products (see Stengel et al., 2017).

AC:

RC: Related to the above [and also to material on p6]: Since CC4CL does keep cloud cover as primary variable, it should be explained whether the thresholds (table 2) vary (for example, with the specific sensor or orbit), or whether they are fixed once and for all, now that they have been optimized via the ANN technique. More importantly, do the weights as established during the ANN learning process vary? Are they a function of orbit, instrument, illumination, surface, topography. . .? Or else, are all of these dependencies incorporated in one single ANN? If so, how are commonly known problems with ANN (such as overfitting) avoided here? Using this cloud masking and thresholding technique, what is the (minimum) cutoff optical thickness, below which cloud are no longer detected? How do optical thickness detection thesholds vary with surface type and sun-sensor geometry?

AR: The thresholds in table 2 have been quantified through iterative optimisation rather than by the ANN technique (p 6, l 68). They are fixed for all sensors and orbits, and thus, as is shown in Table 2, only vary as a function of illumination and surface condition. There are no sensor specific thresholds, but we apply a simple viewing angle correction on the input satellite data. The ANN weights themselves have been trained with NOAA18 data, and we linearly adjusted input data for other satellites to better match NOAA18. The text already states which ancillary data have been used when training (p 6, l 38 - 42), including surface conditions, and also that several ANNs were produced (p 6, l 32 - 34). We are using 3 different ANNs (day, twilight, night), which reduced the overfitting problem mentioned by the reviewer. Also, overfitting was minimised by comparison with an independent test dataset while training.

We did not quantify a cutoff optical thickness as asked by the reviewer. Instead, our approach involved quantifying those threshold values for which the fit between CC4CL and CALIPSO cloud cover is best.

AC: p6, l 65 ff: "The thresholds themselves vary depending on illumination and surface conditions, namely land, sea, and snow/ice cover (Table 2), and were quantified through iterative optimization. They are fixed for all sensors and orbits."

RC: Related to the above [and also to material on p6]: The three elements of the ANN need to be described better. How well is the pseudo-CALIOP optical depth itself estimated with the ANN? Figure 2 illustrates the performance of the cloud mask after ANNCOD has be converted into a binary cloud mask. Since the ANN predicts ANNCOD and not the cloud mask itself, it should be the performance of the ANN with respect to ANNCOD that should be demonstrated here. In this context again: How is overfitting avoided?

AR: It was never the intention of creating a COD retrieval that can also be used to extract a cloud mask. We aim at creating a binary cloud mask. For a COD retrieval, we would have needed to train the full range of CALIPSO COD (approx. 0-15), but we cut off at a COD of 1 (and set all COD > 1 = 1). We assumed that CALIPSO COD values > 1 are clouds that will always be correctly detected by passive sensors. Considering that, we do not think that a comparison of ANNCOD with CALIPSO COD makes sense and thus should not be included here.

How can the non-linearities of radiative transfer be emulated with a single hidden layer?

AR: We agree that the use of at least one more layer could have improved the retrieval. However, in the CC4CL framework we used an IDL based library who does not provide more than 1 hidden layer.

What is the result for ANNCOD for the training data set as opposed to the test data set?

AR: The training dataset is only a small part of the collocation dataset. When training, the dataset was divided by 90/10 percent into a training dataset and a test dataset, i.e. we trained on 90 percent and tested simultaneously on the rest of the data (10 percent). So, the test dataset has only been tested while training. To avoid the overtraining, the training has been stopped when both RMSE (train/test) started to differ.

How is the correction for viewing angle done?

AR: We found that only a part of the whole viewing angle geometry was trained (0-35° out of up to 70° for AVHRR). We created an averaged ANNCOD with respect to each viewing angle and found a cosine-shaped dependency, which we corrected with an empirical cosine function.

$ANNCOD_{corrected} = ANNCOD - ( 1. / 12. * (1. / \cos( satellite\_zenith\_angle * degree\_to\_radians ) - 1. ) )$

How many inputs does the input layer have; what are they?

AR: This depends on the illumination (day/twilight/night) and the availability of channels. We will add the input variables to the text.

AC: p6 l 37: "For the input layer, input variables are surface temperature, snow/ice cover, and the land/sea mask for all three cloud masks. Regarding sensor data, input channels are Ch1, Ch2, Ch5, Ch6, and Ch5-Ch6 for the day ANN, Ch4, Ch5, Ch6, Ch5-Ch4, and Ch5-Ch6 for the night ANN, and Ch5, Ch6, and Ch5-Ch6 for the twilight ANN."

What is the activation function? Are there bias perceptrons?

AR: Our activation function is the sigmoid function. We did include bias perceptrons.

What motivates the use of one single hidden layer, and why are there 50 neurons in it?

AR: Regarding the one hidden layer, see comment above. Will add text an number of neurons.

AC: p6 l37: "Through incremental testing, we found that 50 neurons was the value for which the trade-off between output quality and computing speed was optimal."

Is the network re-trained for every new satellite data set, or are the weights fixed?

AR: The weights are fixed, see p6 l 68 – p7 l 6. We tried to overcome the problem of having different shapes of the spectral response functions by applying linear regression coefficients (see Table 3). In a later version of the cloud mask, we applied a more sophisticated approach through using multispectral observations of IASI and SCIAMACHI. In the version presented here however, the linear regression is based on a one month triple collocation between AVHRR NOAA18, AATSR-ENVISAT, and MODIS AQUA.

How exactly were the threshold values from table 2 determined that are applied to ANNCOD to translate into cloud mask?

AR: We determined these thresholds through incremental application of a skill score analysis of the ANN cloud mask with CALIPSO for the whole collocation dataset (as a reminder, the training dataset is only a small part of the collocation data set). See previous comment, including a text change at p6, l 65 ff.

Finally, what is the quality of the thermodynamic phase retrieval, optical thickness and effective radius, depending on how close ANNCOD is to the cloud detection threshold?

AR: We did not quantify this relationship in detail. However, Figure 6 shows retrieval uncertainties of CTP, COT, and CER together with cloud mask uncertainty. The patterns do not appear to be clearly related. A quantitative analysis, e.g. calculating correlations between relative uncertainties, would certainly provide more detailed answers, but was out of this paper's scope.

AC: p18 l30: "It would also be worth investigating the relationship between the quality of retrieved variables (CTH, COT, CER, cloud phase) and cloud mask uncertainty."

RC: Essentially, the paper claims that a cloud retrieval is attempted if the optical thickness exceeds 0.4 over snow/ice during day light conditions. This would be a remarkable improvement over existing retrievals. MODIS usually does not detect clouds over snow covered areas in the Arctic unless they have an optical thickness significantly larger than 0.4 (around 7). CC4CL would be an improvement of an order of magnitude, and the question is whether the cloud retrievals would be of practical use,

especially when applying them to AVHRR instead of MODIS. The reviewer strongly believes that the only way to achieve detection thresholds on the order of 0.4 in optical thickness in snow/ice covered regions in the Arctic, one would need to use convolutional layers (i.e., use multi-pixel retrieval approaches).

AR: Please note that the ANNCOD is a *pseudo* optical depth. The threshold value, here 0.4 for daytime over snow/ice, is a relative value between 0 and 1. It does not provide any information on the absolute optical thickness value, but is rather a normalized optical thickness that attempts to fit CALIPSO measurements.

RC: p2,l78-80: "Consistency can be traded for continuity" needs clarification. Perhaps this can be done while elaborating on CDR (see comment above). This discussion will contribute to a better motivation of this study.

AR: Agreed, will clarify. See also reviewer #2 comments on the same issue.

AC: "Consistency in approach can be traded for continuity of results, and multi-platform algorithms could exploit additional data when newer sensors become available"

RC: p2,l90: "MODIS provides": is a partial repetition of material in the left column of the same page.

AR: Agreed, will delete that sentence, as it is redundant.

RC: p3,l18: "on other" > "over other"

AR: Will rephrase.

AC: "improvement over other established"

RC: p3,l20: Which "macrophysical" product do the authors have in mind here? What exactly does "radiative inconsistent" mean (supposedly, macroscopical products are inconsistent with microphysical products, but this is different from "radiatively inconsistent"; the reader is currently left to guess here). How exactly does the CC4CL approach ensure radiative consistency amongst all input satellite radiances (and all output products)? Indeed, other approaches have a cloud mask that may be independently derived from the microphysics products. Simply stating that CC4CL is "different" in this regard does not support the statement that it is more "consistent". More details are needed to add specificity.

AR: We are referring to macrophysical products such as CTT and CTH. Please see also our related answer on radiative inconsistency above, and the detailed description of the issue in part II of this paper.

RC: p3,l46: Quantify "very realistic", or just use "realistic"

AR: Will rephrase.

AC: "provides realistic estimates"

RC: p4,l35: Auxiliary > Ancillary

AR: Agreed.

AC: "Ancillary"

RC: p4,l38: Neural Network not yet defined at this point. May need the NN section prior to this statement.

AR: Will add a reference to the ANN section here. Most readers probably have at least a vague idea what a neural network is.

AC: "and as input to a neural network cloud mask (see Section 3.1.1)"

RC: p4,l73/l75: "optimal estimation", "cloud typing scheme". None of these have been described at this point in the manuscript. Sequence needs to be re-shuffled.

AR: Will add section references here. Again, these references should be sufficient, as the readers will have heard these terms before and do not require a detailed definition here to understand the following text.

AC: "The USGS data are used as a land sea mask within the optimal estimation retrieval (Section 3.3.3), as well as a land cover classificator within the cloud mask and the Pavolonis cloud typing scheme (Section 3.3.2)."

RC: p5,l1: "were" > "are"
AR: Cannot find "were" in that sentence.

RC: p5,l1: Reference and/or data source (link) needed for CALIPSO product
AR: Cannot find the CALIPSO product in that sentence. Maybe there is a linenumber mismatch?

RC: p5,l68-l72: multiple acronyms need to be introduced prior to first use.
AR: Most acronyms in that sentence were introduced in the first paragraph on page 2.
AC: "…Clouds from AVHRR Extended (CLAVR-X) (…) Global Cloud and Aerosol Dataset Produced by the Global Retrieval of ATSR Cloud Parameters and Evaluation (GRAPE) …"

RC: p5,l79: The outcomes of the study should be at least summarized here. Also, the use of "round robin" may not be ideal for an international readership as it is a cultural reference (British/American) that may not be commonly known. Consider paraphrasing the technique instead.
AR: Will rephrase.
AC: "was chosen from three competing algorithms in a "round-robin" (i.e. each algorithm is tested against all other algorithms) analysis. All algorithms have proven their maturity for deriving the considered cloud parameters (cloud cover, liquid and ice water path, cloud top height) from AVHRR and MODIS data (Stengel et al., 2015)."

RC: p5,l98: Do these channel numbers refer to the CC4CL IDs from table 1?
AR: Yes, will clarify.
AC: "The albedo of snow/ice covered pixels is set to globally constant values of 0.958 (Ch1, CC4CL ID as in Table 1), 0.868 (Ch2), 0.0364 (Ch3), and 0.0 (Ch4),"

RC: p6: Cloud detection: See multiple comments above (following p2,l55-l57 comment). Also: Are there any convolutional layers included in the approach? This would have allowed capitalizing on the context of a pixel.
AR: We did not add any convolutional layers in the ANN. See above comments regarding cloud detection and the ANN.

RC: p7,table 3: How was the regression done - based on radiance or irradiance, based on counts? Based on brightness temperature (for IR channels)? The offsets seem rather large; what is the explanation for significant offsets?
AR: The regression coefficients were calculated based on reflectance and brightness temperature data. The offsets might be a result of imperfect collocation, relative calibration differences, and mainly differences in spectral response functions. It is difficult to quantify the contribution of each, but spectral response probably explains most of the offset.

RC: p7,l49: VIIRS algorithm is used: What is the purpose of this statement? If it is kept, this needs to be elaborated (what does the VIIRS algorithm do differently). Also, there are various other algorithms that are improved over the heritage algorithms, which would probably all need to be mentioned here (or at least a subset thereof).
AR: This paragraph is a brief summary of the Pavalonis algorithm performance. Not surprisingly, it performs better if more spectral channels are used for cloud typing. The sentence emphasizes the generic limitation of using only AVHRR heritage channels, which does not only affect cloud detection or optimal estimation, but also cloud typing. We do not think that other algorithms using other data than the heritage algorithms need to be elaborated or mentioned here (however, we do so elsewhere). It is obvious that they perform better if using more channels, but that is not the point here.

RC: Figure 2: This is just one example where labels are too small, and are too pixelated. Generally improve the figure quality and enlarge labels. About the content: It is rather hard to interpret this figure. The x-axis is "normalized". Does that mean that the difference of the ANNCOD-retrieved value and the threshold from table 2 is divided by the threshold value itself? Does "x=0" mean that the retrieved optical thickness equals the threshold per table 2? Does the "CLEAR" label refer to CALIPSO? For x=-0.2, we find an uncertainty of 40%. Does that mean that CC4CL misclassifies clear pixels as "cloudy" in 40% of cases?

AR: As mentioned in the text, the x-axis is normalized, i.e. the difference between ANNCOD and the threshold was divided by the threshold. Yes, x=0 means no difference between ANNCOD and the threshold. Again, please remember that this is a *pseudo* optical thickness. CLEAR means that the ANN cloud mask defined a pixel as cloud free. It shows that we need different equations to quantify uncertainty for clear and cloudy cases. The text also explains how uncertainty is calculated: 100 – PEC [%], with PEC = the ratio between all correctly classified pixels and the number of all pixels analysed. Also, if x=-0.2, CC4CL misclassifies cloudy pixel as 'clear' in 40% of the cases with respect to CALIPSO. The uncertainty defines the misclassification of CC4CL compared to CALIPSO for a certain combination of ANNCOD and the threshold used.

See below Figure 2 with larger labels and annotations. We also increased labels for Figures 3-7 (see responses to other reviewers).

[Figure]

RC: p8,l70: Why are largest uncertainties found for opaque clouds? Also, figure 10 does not show quantitative evidence for this statement - colors are harder to interpret than numbers on a graph. Can this somewhat counterintuitive statement be supported by a more succinct graph?

AR: We are referencing the wrong figures. Will correct.

AC: "COT uncertainties increase with COT magnitude, and largest uncertainties are found in cases of opaque cloud coverage (Figure 4 middle and Figure 6 topright)."

RC: p9,l3-5: :Validation is show for . . . rather than: Unclear. What is the difference between CTH and "its" retrieved value?

AR: CTH is a derived variable, i.e. derived from CTP, which is the retrieved value. Will clarify.

AC: "The validation is shown for comparisons of CTH (derived from CTP) rather than CTP (retrieved) to enable…"

RC: p9,l13: "TOA radiation is the *sum total* of emission and scattering throughout the atmospheric column" - please formulate this more accurately: What is a "sum total" of two processes? Also, the next paragraph more or less paraphrases Platnick's vertical weighting function paper where this is formulated more accurately, and where the concept of a weighting function is well explained. Please cite that paper and use similar terminology here. As for multi-layer clouds, there is a fairly new paper by Wind, Platnick et al. (http://journals.ametsoc.org/doi/abs/10.1175/2010JAMC2364.1), but it is probably not applicable to this paper here because of the channel selection.
AR: Will clarify.
AC: "However, TOA radiation is the product of emission and scattering processes throughout the atmospheric column (Platnick, 2000)."

Platnick, S. (2000), Vertical photon transport in cloud remote sensing problems, J. Geophys. Res., 105(D18), 22919–22935, doi:10.1029/2000JD900333.

RC: p10,l1: How is the CTH adjustment done if the cloud base is not known? Where does cloud base (or cloud geometrical thickness) information come from?
AR: We approximate the observed temperature as emitted from one optical depth into the cloud. Assuming the cloud is vertically homogeneous with a constant lapse rate $\Gamma$, we can write the thickness-corrected CTT as,
$T_{cor} = BT(\lambda) + \Gamma / (\sigma N)$,
where BT is the observed brightness temperature, $\sigma$ is the cloud particle cross-section, and N is the cloud particle number concentration. Using the observations at 11µm and 12µm provides two simultaneous equations in $T_{cor}$ which can be solved, using $\sigma$ values for a LUT.

RC: p10,l9: Does this statement about sectors refer to figure 9? Please match figures and text, otherwise figures become "orphans" that are not tied to the manuscript.
AR: Will add figure references.
AC: "CC4CL correctly classifies all pixels as cloud covered, with a few exceptions in sectors 3 and 4 (Figures 8 and 9)."

RC: p10,l14: Please define what is meant by "surface" in this case.
AR: Will clarify.
AC: "In the case of a (semi-)transparent cloud top layer, multiple surfaces (several cloud layers, Earth surface) contribute to the observed satellite data."

RC: p10,l16: insert "a" before "single-layer"
AR: Agreed.
AC: "For a single layer, optically thick (COT > 1) cloud,…"

RC: Figure 7: please enlarge labels, as well as histograms; it is hard to compare the retrievals quantitatively otherwise. Also: It would really help if histograms were shown separately for snow-covered areas as opposed to dark surfaces. It is expected that retrieval quality would differ significantly depending on the surface conditions.
AR: We will enlarge labels and histograms. However, we do think it is sufficient to show histograms for all surfaces combined to make our point that there are differences between retrievals, which is also supported by the statistics.

[Figure]

[Figure]

[Figure]

RC: p12,l41: "performance of existing algorithms" What are the "existing algorithms" that CC4CL? Has the manuscript shown that these existing algorithms perform less well than CC4CL.
AR: Will remove the subordinate clause.
AC: "In general, the quantitative and qualitative agreement between CC4CL and CALIOP CTH is impressive."

RC: p12,l88: "AVHRR" > "for AVHRR"
AR: Agreed.
AC: "and AATSR data than for AVHRR"

RC: p12,l89: Should "continually" be replaced with "consistently"? Unclear what this statement means. If it were "consistently" it would be more clear, but the word order should be fixed: "The CC4CL phase identification does not agree with any of the three CALIOP cloud flags consistently, which is reasonable given . . ."
AR: Agreed, we will rephrase as suggested.
AC: "The CC4CL phase identification does not agree with any of the three CALIOP cloud flags consistently, which is reasonable given . . ."

RC: p13,l19/20: ". . .insensitive to the specific instrument evaluated, such that the merged data set is sensible". What does this statement mean? The paper does not actually present a *merged* data set, or was that the actual intent of the paper? It does evaluate collocated overpasses from different satellites, but these are not merged in the sense of a CDR. Please remove the statement about "merging" data sets unless this was the actual intent of the paper (in which case it would need to be modified considerably).
AR: We will remove the statement about "merging" datasets, which was once foreseen in the project but has not been done at the moment this paper was written.
AC: "In general, the retrieved values are insensitive to the specific instrument evaluated. Absolute…"

RC: p13,l31: "disagree nonetheless": They disagree despite their channels are fairly close? Can this be re-phrased? The whole paragraph is a bit roundabout. There's a 30-40% difference in reflectance, but "their" retrieval values are "much more similar"? Please make this statement more precise. "The difference to AVHRR and MODIS is largest for CER" - does this statement refer to AATSR again?
AR: Will rephrase, as these statements are definitively hard to understand. Yes, the last sentence refers to AATSR.
AC: "Also, even though spectral response differences are largest between MODIS and AVHRR (which results in a reflectance difference of up to 30–40 % (Trishchenko et al., 2002)), their retrieval values are much more similar. The difference between AATSR and both AVHRR and MODIS is largest for CER, so microphysical variables, which are derived from reflectance data only, appear to be most affected."

RC: p13,l39: The t-test needs to be explained in much more detail. What is H0, what is mu1, what is mu2? Are we talking about the covariance between two data sets, which is assessed using the t-test approach? If so, are the data from the two different data sets (supposedly this is what "mu1" and

"mu2" refer to) re-gridded to one common grid before comparing them? The premise of this statement deserves at least one paragraph, if not half a page.

AR: This is a very basic t-test, using a well-defined symbology. It is a test for significance of the difference between the mean values of two populations (i.e. $\mu 1$ = mean of population 1, $\mu 2$ = mean of population 2). The data were indeed re-gridded to a common grid, which is all explained in section 2.3.

AC: "The differences between mean values ($\mu 1$ and $\mu 2$) are almost always significant (t-Test p-value $< 0.1$, $H_0$: $\mu 1 = \mu 2$)."

RC: p13,l45: "spatiotemporally collocated sensors": The sensors are not collocated - is that the point of the statement? Or is this an explanation why the t-test "fails? What does "non-significant" t-test mean? Could the strictness of the comparison be relaxed by gridding the retrievals to a coarser common grid before making the inter-comparison?

AR: The sensors are not collocated, but the data are. And the collocation should minimize differences due to observation times and observation area. The significance level is now mentioned in the correction above, but can also be found in the caption of Table 6. As said above, the data were re-gridded.

AC: "…when driven with spatiotemporally collocated satellite data obtained from three different sensors."

RC: p13,l57: "depending on the user's application" - this needs to be clarified. For which applications can they be used interchangeably? Could a combined AVHRR and MODIS cloud data record constitute a CDR (would it meet the requirements)? As stated above, the manuscript does not actually "merge" data sets in this way, but more specificity would be helpful here.

AR: We added references to give examples. We do not think that the AVHRR and MODIS cloud data record should be seen as one continuous, consistent data record. Rather, AVHRR provides the opportunity of long-term data coverage back to 1982, providing data that are at least comparable to MODIS. That certainly excludes local analyses, but rather refers to continental to global applications.

AC: "depending on the user's application, such as model validation, data assimilation applications, or climate studies in general (Liu et al., 2017, Yang et al., 2016)."

Liu, C., R. P. Allan, M. Mayer, P. Hyder, N. G. Loeb, C. D. Roberts, M. Valdivieso, J. M. Edwards, and P.-L. Vidale (2017), Evaluation of satellite and reanalysis-based global net surface energy flux and uncertainty estimates, J. Geophys. Res. Atmos., 122, 6250–6272, doi:10.1002/2017JD026616.

Yang, Qinghua, et al. "Brief communication: The challenge and benefit of using sea ice concentration satellite data products with uncertainty estimates in summer sea ice data assimilation." The Cryosphere, vol. 10, no. 2, 2016, p. 761.

RC: p13,l77: "we see that COT uncertainty scales with COT itself": this is not shown in the manuscript. If it is, please refer to a figure or section.

AR: As mentioned above, it is shown in Figure 4 middle and Figure 6 topright.

AC: "we see that COT uncertainty scales with COT itself (Figure 4 middle and Figure 6 topright)"

RC: p13,l79-l88: Consider re-writing this section; simplify and use literature references; most of these observations have been documented before (large COT uncertainty as reflectance approaches asymptotic value; large uncertainties for bright surfaces).

AR: Will simplify and add references.

AC: "CC4CL COT values are at times unnaturally large, and the associated uncertainty reflects that. Also, it highlights under which conditions the optimal estimator converges to a solution with a relatively large divergence from the measurements, which here are associated with optically thick clouds or underlying snow/ice cover (see also Kahn et al., 2015, Wang et al., 2011). COT and CER

uncertainties are clearly largest, and reflect the limited information available with which to retrieve these values. For further possible explanations due to assumptions and limitations within the methodology applied, please see part II."

Kahn, B. H., M. M. Schreier, Q. Yue, E. J. Fetzer, F. W. Irion, S. Platnick, C. Wang, S. L. Nasiri, and T. S. L'Ecuyer (2015), Pixel-scale assessment and uncertainty analysis of AIRS and MODIS ice cloud optical thickness and effective radius, J. Geophys. Res. Atmos., 120, 11,669–11,689, doi:10.1002/2015JD023950.

Wang, C., P. Yang, B.A. Baum, S. Platnick, A.K. Heidinger, Y. Hu, and R.E. Holz, 2011: Retrieval of Ice Cloud Optical Thickness and Effective Particle Size Using a Fast Infrared Radiative Transfer Model. J. Appl. Meteor. Climatol., 50, 2283–2297, https://doi.org/10.1175/JAMC-D-11-067.1

RC: p15, l11: "otherwise are" > "otherwise they are"
AR: Will rephrase.
AC: "otherwise they are"

RC: p15,l15: "may it stem" does not work in English; consider "whether it stems from. . . or"
AR: Will rephrase.
AC: "whether it stems from a cloud or the Earth's surface"

RC: p15,figure 11: The table below the cross section is too small. Also, what happened at lat=61? Why do the active imagers pick up a cloud where CALIPSO does not?
AR: Unfortunately, the table itself cannot be increased due to space limitations and a bug in the Python library applied to produce the table. The colours show cloud phase. Cloud type numbers are not as important, and we could have removed them as for Figure 15. At latitude 61°, we see that there are broken cloud fields in the area, which might have appeared in the sensor's field of view but not in CALIPSO's.

RC: p16,l8: consider "a conscious decision was made to [deliberately] trade. . ."
AR: Will rephrase.
AC: "For ESA Cloud_cci, a conscious decision was made to trade spectral information for time series continuity."

RC: p16,l19: "on a first view" > "at first glance"
AR: Will rephrase.
AC: "At first glance, estimates of…"

RC: p18,l59: "synergic" > "synergistic"
AR: Will rephrase.
AC: "exploits synergistic capabilities of several EO missions"

RC: p18,l95: "accurate and precise": These two were not discussed separately. Where was this done? If not, please clarify this statement.
AR: We will remove precise, which stands for a low standard deviation of errors (not shown here). The results are accurate due to the relatively low bias.
AC: "optically thick cloud retrievals are very accurate when compared against CALIOP (bias < 240 m)"

---

## Author Comment (AC2) · 5 Mar 2018

RC: referee comment

AR: author response

AC: author's changes in manuscript
* * *
Review of "The Community Cloud retrieval for Climate (CC4CL). Part I: A framework applied to multiple satellite imaging sensors", by Sus et al.

RC: The manuscript introduces a valuable approach to establish a common passive cloud retrieval applicable to a series of standard polar orbiters in order to create data sets usable for climatological studies. This would be an important step for the community and the usability of satellite products outside the satellite community. I also understand and acknowledge the need to base such an approach on well established methods instead of more experimental approaches as suggested by one of the other referees. The general presentation is of good/excellent quality. My two co-referees have elaborated on a number of specific technical and scientific details already. I want to focus on a more general weakness.

What exactly is the focus of this manuscript? If I missed important, clear, early statements in the existing text, I apologize. If not, the reader needs this guideline. In many places important details can not be given and are not explained owed to the shear extent of this project. In most cases the reader is then correctly referred to other publications where the methods of CC4CL are introduced. This way the purpose of the manuscript at hand becomes more and more unclear while reading through it. First impression is that the general method will be explained. But then the core retrieval techniques are explained elsewhere (McGarragh). Then a technical explanation of the ANN cloud mask is started, but it stays too short to be fully comprehensible. After the introduction of example cases Fig 3-5 and cross sections Fig 9-16, I expected an in-depth discussion of reason for differences and a quantitative validation (section titles containing "validation") or cross-comparison of all products, but the discussion stays very general and mostly describes differences. Proper validation is again shown elsewhere (Stengel).

The limited original content of this manuscript (correct me, if I'm wrong) is not reflected by the title and manuscript length (e.g. 8 figures 8-15 with very comparable content and not too surprising differences between active and passive sensor, but no quantitative validation). The authors should clarify the purpose of this manuscript and shorten parts published elsewhere even stronger. I suggest to consider these general points and a revision of the manuscript.

AR: We appreciate the comments of referee 3 and agree that some clarification is required to explain the purpose of the paper. Please note that this has also been pointed out by referee #2, so our answer here has been copied from our comments to reviewer #2.

This paper's main purpose is to present a new cloud retrieval framework (CC4CL). It is a two part publication that contains a detailed description of the retrieval algorithm in part II. Part I should not be seen as a validation paper, but rather contains a section that provides the reader with an overview of the functionality of CC4CL, including generic strengths and weaknesses. The goal is to inform the reader of potential applications of this data in future research. The four case studies aim to illustrate the strengths and weaknesses of CC4CL through detailed, direct (i.e. with very little averaging), and collocated comparisons with independent CALIOP data. The Stengel paper, as the reviewer correctly mentions, contains a true validation of CC4CL, but to include such an in-depth analysis here would have substantially increased the paper's length. We think that keeping part I concise and focused better serves its purpose as an introduction to the functionality and generic applicability of CC4CL. For readers who might be interested in a validation of CC4CL after reading part I, we refer to the Stengel paper in the text.

However, we will replace "validation" with "examination" or "analysis" throughout the text. The reviewer is correct that no true validation study has been carried out here, and we rephrase in order to avoid misunderstandings.

Specific major issues:

RC: p3, line 27: "Moreover, the resulting time series are carefully validated ... (ISCCP, PATMOS-x, CM SAF, and MODIS Collection 6), reanalysis and model data (ERAInterim and EC-Earth), ground-truth synoptic observations, and CALIOP lidar data."

My understanding was that I would see that in this manuscript: You will only show CALIOP comparisons, will you? Could you please clarify.
AR: Yes, we only compared with CALIOP. We will add a reference here to the Stengel paper, and also a reference to our internal product validation report.
AC: "Moreover, the resulting time series were carefully validated against well-established climatologies (ISCCP, PATMOS-x, CM SAF, and MODIS Collection 6), reanalysis and model data (ERAInterim and EC-Earth), ground-truth synoptic observations, and CALIOP lidar data (Stengel et. Al, 2017, PVIR)."

RC: p8, section 4.3: I think you cannot call this chapter "validation". There is no systematic validation, only a few selected case studies, which mainly show the problems and no systematic quantitative validation. Four case studies of time height cross sections are shown only to present that lidar cth does not have much to do with passive cth? I also expected CER and COT validation somewhere.
AR: We agree with the reviewer and will, as mentioned above, replace "validation" with "comparison". As the reviewer mentions, this comparison shows the generic strengths and weaknesses of CC4CL, which certainly relates to the processing of passive imager data. However, the reader should appreciate the basic functionality of CC4CL and we find that these local comparisons are well suited for that purpose. Please also note that the CALIOP COT information is less reliable than CTH, which is why we did not compare with COT.
AC: "Comparison with CALIOP"

RC: p13, line 15: You mean, proper quantitative validation is shown in another paper ...Stengel et al 2017 ESSD? The retrieval method was shown in two other papers as well... McGarragh 2017 at JAS and AMT. Remind me about the reason for this manuscript?
AR: Please see our comments above.

Minor issues:

RC: p 2, line 33: AVHRR was not introduced before, was it?
AR: The reviewer is correct. Will rephrase.
AC: "Compared to the Advanced Very High Resolution Radiometer (AVHRR), MODIS has several…"

RC: p 2, line 53: What is r?
AR: The Pearson correlation coefficient.
AC: "(up to a Pearson correlation coefficient r = 0.94)"

RC: p 2, line 65: How can CTP and CTH be underestimated at the same time? Can you please comment?
AR: We agree that the use of the word "underestimates" twice suggests that CLARA-A2 is wrong in both cases. However, whereas CALIOP data are considered to be "truth" data, we will now simply state that CLARA-A2 has a *lower* CTP than the other retrievals, which is not an underestimation, just a different retrieval outcome.
AC: "Comparing CLARA-A2 to PATMOS-X, MODIS C6 and ISCCP, global CTP is lower by 4–90 hPa…"

RC: p 2, line 66: What is a "cloud phase bias ... of 9%"? Cloud phase? Liquid and ice? Or cloud cover?
AR: This refers to the fraction of liquid clouds.

RC: p 2, line 68: Low or high bias?
AR: We will specify.
AC: "+ 197 m"

RC: p 2, line 102: It would be nice to say at this early stage what the purpose of this particular manuscript is in ESA Cloud_cci? And what other parallel publications contribute? Later on, the reader gets the impression that everything relevant is introduced elsewhere.
AR: We agree that this needs clarification. We copied our answer to reviewer #1, who made a similar comment.
AC: "The European Space Agency has established the ESA Climate Change Initiative program (ESA CCI, 2015; Hollmann et al., 2013) in order to advance knowledge of the climate system through the generation of satellite based data records utilizing European and non-European assets. The CCI project's primary focus is the production of thirteen Essential Climate Variables (ECVs) covering ocean, atmospheric, and land geophysical variables. With these data records CCI is aiming to fulfil highest climate requirements from the Global Climate Observing System (GCOS). The study presented here is part of the ESA CCI for clouds (ESA Cloud_cci), which has the objective to develop a state-of-the-art open-source community cloud retrieval algorithm being capable of processing passive satellite imager data for several decades. Both in part I and part II of this paper, we present the processing framework as developed within ESA Cloud_cci (CC4CL, part I), the detailed mechanisms of the optimal estimation retrieval (part II), and provide an initial assessment of the strengths and weaknesses of derived cloud parameters (part I). With CC4CL several decades of passive imaging satellite data have been processed and are made available to the user. The resulting climate data records (CDR) are presented in Stengel et al., 2017."

RC: p 6, line 27: If this is the only description of ANNCOD available, you might at least want to cite Kox et al . 2014 (AMT, 7, doi:10.5194/amt-7-3233-2014) who introduced the idea and described in much more detail.
AR: Please also our answers to reviewer #1, who asked for a more detailed introduction of ANNCOD, which we will provide. Kox et al. developed an approach similar to ours for retrieving Cirrus COT and CTH, but we do not think that they introduced our idea for cloud masking.

RC: p6, line 51ff: This is all a slightly vague description, if it isn't detailed somewhere else. Why do you need ... after viewing angle dependency correction ... a whole set of thresholds? ANNCOD already gives an answer on the question cloud or no-cloud, doesn't it?

AR: That would mean that the ANNCOD perfectly reproduces CALIPSO data, which is not the case. The thresholds were necessary to avoid overestimation of cloud cover due to the sensitivity of the passive sensors. With the passive sensor we measure reflectance and temperatures, in contrast to CALIPSO which is independent of both. Strongly reflecting surfaces and/or difficult illumination conditions will create ambiguities. Especially under difficult illumination conditions such as twilight, and over ice/snow surfaces we needed to increase the thresholds to avoid overestimation and decrease the false alarm rate (knowing that we might miss some clouds). We made a skill analysis with CALIOP to find the most suitable thresholds. The viewing angle correction has nothing to do with this, but more or less you can see this as a sun-zenith and surface correction of the retrieval.

RC: p7, Figure 2: y-axis. It is PEC not 1-PEC shown, isn't it? Does the graph show that, at your threshold you are only correct by about 50%?? Please discuss.

AR: The y-axis shows 100 – PEC [%]. The graph shows that the uncertainty increases to about 50 % at the threshold. This makes sense, as an ANNCOD value close to its threshold indicates that no clear distinction between cloud/no cloud can be made, thus the highest uncertainty. The larger the difference between ANNCOD and its threshold, the lower the associated cloud mask uncertainty.

RC: p10, line 10: "consistent". You could also say its all over the place, with different physical reasons in any single column. This is not a validation. You even tried to correct cth for cc4cl and still have big problems.

AR: We do find that Figure 9 shows very similar retrieval results of CTH for all three sensors, except in sector 2. We are referring here to the agreement *amongst* sensors, not between sensors and CALIOP data.

RC: p11, Figure 7: Please make the labels consistent with the rest of the manuscript: n18->avhrr, myd->modis ...

AR: We will modify labels accordingly.

---

## Author Comment (AC3) · 5 Mar 2018

RC: referee comment

AR: author response

AC: author's changes in manuscript
* * *
RC: p 2, line 5: I would add cloud forward model assumptions to the list of secondary confounding factors

AR: Will add „cloud forward model assumptions" to list.

AC: „Several secondary variables (cloud forward model assumptions, state of surface and atmosphere, viewing geometry, sensor calibration and spectral response uncertainties) …"

RC: p 2, lines 12-14: The CERES-MODIS products (e.g., Minnis et al., 2011a,b, IEEE TGRS) should also be included here.

AR: Will add the CERES-MODIS products.

AC: „and MODIS Collection 6 (MODIS C6) (Platnick et al., 2017) as well as the CERES-MODIS products (Minnis et al., 2011)."

P. Minnis *et al.*, "CERES Edition-2 Cloud Property Retrievals Using TRMM VIRS and Terra and Aqua MODIS Data—Part I: Algorithms," in *IEEE Transactions on Geoscience and Remote Sensing*, vol. 49, no. 11, pp. 4374-4400, Nov. 2011.

RC: p 2, lines 23-24: The MODIS C6 phase referred to here is the IR phase of Baum et al. (2012), which is in fact a quad-spectral algorithm (7.3, 8.5, 11, 12µm channels) using β ratios (the authors' description is more appropriate for the C5 algorithm). This IR phase algorithm is run in conjunction with, and is informed by, the cloud top property retrieval algorithm. The authors should be aware, and I believe that they are given the reference to Marchant et al. (2016) later in the paper, that this IR algorithm does not determine phase for the C6 cloud optical properties retrieval; phase for the optical retrieval is determined by the Marchant algorithm that uses the IR phase as one piece of information. Results from the IR and cloud optical properties phase algorithms are often at odds, specifically in cases where phase is more ambiguous.

AR: Will correct the text to reflect that the Marchant algorithm is applied.

AC: ", or a majority vote algorithm that combines four phase tests based on CTT, tri-spectral IR, 1.38 µm, and spectral CER data (Marchant et al., 2016)."

RC: p 2, line 24: Should probably specify that the additional spectral channels are at shortwave infrared (SWIR) wavelengths.

AR: Will add SWIR here.

AC: ",… MODIS has several additional spectral channels at shortwave infrared (SWIR) wavelengths that provide…"

RC: p 2, lines 29-30: Indeed, this is an inherent limitation of the spectral information content of passive IR channels!

AR: Will rephrase.

AC: "these studies show that current retrievals underestimate cloud top pressure for optically thin clouds due to the inherent limitation of the spectral information content of passive IR channels."

RC: p 2, lines 31-35: I assume from the references given that cloud cover refers to cloud fraction or related metrics, and not to geophysical retrievals.

AR: Yes, we are referring to cloud fraction. Will replace cloud cover with cloud fraction.

AC: "There are numerous studies that evaluate the performance of the aforementioned retrievals for cloud fraction with weather (…).  More importantly, these studies emphasize the difficulty of deriving reliable cloud fraction trends from AVHRR time series, as the retrievals overestimate the change in cloud fraction by as much as an order of magnitude"

RC: p 3, line 5: Is the cloud phase bias positive or negative?

AR: The cited bias values were reported as absolute numbers.

AC: "and has an absolute cloud phase bias of lower than + 9 %"

RC: p 3, lines 6-7: See my p 2 comment above regarding MODIS phase algorithms; this statement again refers only to the IR phase.

AR: Will rephrase to refer to Marchant et al., 2016.

AC: "and the phase detection has been improved for liquid clouds. However, the detection of optically thin ice clouds over warm, bright surfaces remains problematic (Marchant et al., 2016)."

RC: p 3, lines 10-11: What is the difference between consistency and continuity? I can surmise that it is consistency in approach versus continuity of results, but it is not clear to the general reader.

AR: The reviewer's assumption is correct, will clarify.

AC: "Consistency in approach can be traded for continuity of results, and multi-platform algorithms could exploit additional data when newer sensors become available"

RC: p 3, lines 34-35: It's not initially clear why independent retrievals of COT/CER and macrophysical products are inherently radiatively inconsistent. I would guess that it depends on the

approach, i.e., how (or if) one set of retrievals informs the retrieval of the other. Can the authors better explain?

AR: The effect of COT/CER/CTH on the top-of-atmosphere (TOA) radiances differs between the different sensing bands as a function of atmospheric state. For example if you used just the 11 or 12 micron measurement to estimate CTH then you must assume something about the COT (usually that it is thick) and something about the CER (typically a climatological value). If the COT assumption is incorrect (e.g. cloud is not thick) so that more upward radiance is transmitted through the cloud than expected, then the cloud top appears too warm and is located (incorrectly) lower in the atmosphere. On the other hand using an all channel fit, as we did here, will identify the cloud as optically thin (from the visible and near visible reflectance measurements) and will avoid this error.

We note that retrieving a specific cloud property from a specific channel is radiatively inconsistent (as example above) but it is generally possible to do a sequential optimal estimation retrieval. In this case one iterates through the channels improving the estimates of CTH/CER/COT with each step. The final result should be the same as an all channel optimal retrieval. This method is not adopted for our problem as it would be computationally less efficient.

AC: "but macrophysical products are estimated independently and are thus radiatively inconsistent with the former variables. Here, parameters are retrieved simultaneously, providing a retrieval that is radiatively consistent over the wavelengths of the observations, given that the instrument's noise characteristics are well known."

RC: p 4, line 1: Retrieval uncertainty estimates that propagate errors is not a novel feature of CC4CL. See, for instance, the MODIS C6 cloud optical properties (Platnick et al., 2017), which provide pixel-level retrieval uncertainties calculated in a manner that is mathematically consistent with that of optimal estimation (although the uncertainties are not part of the solution process).

AR: Agreed, will clarify.

AC: "Another key feature of CC4CL is the production of uncertainty estimates of retrieval parameters (see also Platnick et al., 2017) through explicit error propagation from input to output data."

RC: p 4, line 6: Following on my comment above, neither the optimal estimation approach nor the uncertainty quantification are novel features of CC4CL. As the authors themselves state on p 2, PATMOS-x uses optimal estimation theory, and the MODIS C6 (and C5) cloud optical properties provide rigorous pixel-level uncertainties.

AR: Agreed, will remove novel here.

AC: "We particularly focus on discussing the key features of the framework: the optimal estimation approach in general, …"

RC: p 4, lines 5-13: Regarding statements about consistency of the long-term, multiplatform time series, and the potential of the framework for climate studies, I don't think the authors make a convincing case for either in the text that follows. Four case studies hardly constitute a "comprehensive and detailed analysis of retrieval results," and certainly do not provide enough evidence of the potential for climate studies. Such statements require detailed analyses of long-term and large-scale inter-sensor statistical comparisons, which it appears are actually presented in a companion paper in a different journal (Stengel et al., 2017). It's thus not clear to me why the present paper was not instead a part of the Stengel paper, or vice versa. Given that the primary contributions

are a brief discussion of the ancillary and data sources and a rather limited CTH analysis, I'm not convinced that this paper can or should stand on its own.

AR: This paper's main purpose is to present a new cloud retrieval framework (CC4CL). It is a two part publication that contains a detailed description of the retrieval algorithm in part II. Part I should not be seen as a validation paper, but rather contains a section that provides the reader with an overview of the functionality of CC4CL, including generic strengths and weaknesses. The goal is to inform the reader of potential applications of this data in future research. The four case studies aim to illustrate the strengths and weaknesses of CC4CL through detailed, direct (i.e. with very little averaging), and collocated comparisons with independent CALIOP data. The Stengel paper, as the reviewer correctly mentions, contains a true validation of CC4CL, but to include such an in-depth analysis here would have substantially increased the paper's length. We think that keeping part I concise and focused better serves its purpose as an introduction to the functionality and generic applicability of CC4CL. For readers who might be interested in a validation of CC4CL after reading part I, we refer to the Stengel paper in the text.

However, we will replace "validated" with "examined", as the former indeed suggests more than the paper intends to provide, and remove "comprehensive".

AC: p 4, line 10: "These are initially examined in a detailed analysis of …"

RC: p 4, line 15: Consider using Level-1 instead of L1, which for some readers implies a Lagrange point 1 orbit.

AR: Will clarify here that L1 stands for Level-1. L1 is standard terminology in this field.

AC: "Level-1 (L1) satellite data"

RC: p 4, lines 21-25: Yes, replacing any AVHRR once its successor becomes available will lessen the impacts of orbital drift (and thus sampling times), but drift impacts are likely still to exist. Are these accounted for in CC4CL, specifically when constructing long-term multi-sensor time series?

AR: Orbital drift effects are not accounted for within CC4CL, which is why we write to only reduce drift-induced changes, not to eliminate them.

RC: p 4, line 29: Regarding filtering channel 3b data, is this to include or exclude that channel?

AR: The filter removes noise artefacts from channel 3b data, which are used in the retrieval.

RC: p 5, lines 8-10: It should be NASA Goddard Space Flight Center.

AR: Will change text.

AC: "the NASA Goddard Space Flight Center performed"

RC: p 5, lines 21-23: "Self-calibrating" is I think a little misleading. MODIS, for instance, has a similar design (onboard black bodies and solar diffuser), yet requires a continual effort to monitor instrument stability and identify/correct calibration drifts, typically using fixed ground targets among others.

AR: Will clarify.

AC: "ATSR is equipped with on-board calibration capabilities, such as two black-body targets for the thermal channels and a sun-illuminated opal target for the visible/near-infrared channels."

RC: p 7, lines 3-4: Has the "gap filling" of the MCD43C1 data been validated? Is the approach similar to what is used in the MCD43B3 gap-filled product (Schaaf et al., 2011, "Aqua and Terra MODIS albedo and reflectance anisotropy products," in Land Remote Sensing and Global Environmental Change: NASA's Earth Observing System and the Science of ASTER and MODIS)?

AR: We did not validate the "gap filling", for which we applied a very basic approach to meet our requirements. The approach applied to gap-fill MCD43B3 data is certainly more sophisticated, but its application in our study was out of scope.

RC: p 6-7, Sections 2.2.3-2.2.4: Have the authors verified that there are not any trends in the land surface BRDF and emissivity time series during the MODIS era? If there are, wouldn't the use of the climatology derived from all MODIS data introduce a discontinuity in the surface time series?

AR: We did not perform a trend analysis for these time series. We agree that a trend in the input data would indeed add an artefact to our retrieval output.

AC: p 7 l 14: "Note that the use of a climatology would add a discontinuity in the surface time series if there are trends in the surface BRDF and emissivity time series during the MODIS era."

RC: p 7, lines 6-7: I disagree that the surface is a minor component of the observed signal, specifically for optically thinner clouds. Thus not accounting for the spectral response functions can introduce biases, particularly in spectral regions such as the near-IR (e.g., AVHRR channel 2, MODIS channel 2) where reflectance by vegetation can change rapidly.

AR: Agreed, will clarify.

AC: "in spectral response functions. Note that this might result in retrieval biases, particularly in spectral regions that are sensitive to rapidly changing environmental processes such as vegetation growth (near-IR)."

RC: p 7, line 16: Resampled or aggregated?

AR: Resampling is defined as the technique of manipulating a digital image and transforming it into another form. Thus the term is applicable here. As is aggregated.

RC: p 7, line 16-17: I would agree that differences in sensor spatial resolution are reduced when averaging radiances/reflectances. However, this is likely not the case when averaging L2 geophysical parameters, as is done here, since the retrievals can have significantly different PDFs within a grid box due to pixel size differences alone.

AR: Agreed, will clarify.

AC: "This resampling is required for an intercomparison of CC4CL Level-2 data on a common grid. However, note that differences in sensor spatial resolution can lead to significantly different PDFs within a grid box, the effect of which we did not analyse."

RC: p 9, line 5: How much data was used to train the ANN? Was an observation time difference filter applied to the NOAA-18/CALIOP co-location?

AR: See p 9, line 10-11. Yes, the time difference filter was 15 minutes.

RC: p 9, lines 19-21: If I understand correctly, the reflectances/radiances were adjusted to account for spectral response differences? Were the co-located observations filtered for cases in which both satellites viewed the scene at the same sun-view angle geometry? Such angle matching is important when comparing solar channels where reflectance is strongly angularly dependent.

AR: Yes, we did account for sun-view angle geometry differences. We filtered all collocations with differences in satellite zenith angle > 0.5°, sun zenith angle > 1°, and observation time > 30 mins.

AC: "We derived appropriate coefficients through linear regression analysis between collocated satellite observations for each input channel pair (Table 03), applying a filter on differences in satellite zenith angle (> 0.5°), sun zenith angle (> 1°), and observation time (> 30 mins)."

RC: p 10, lines 21-22: My understanding is that the uncertainty obtained from the optimal estimation framework can be thought of as the sensitivity of the solution space at the point of the solution to the measurement uncertainty (which includes instrument, ancillary, etc., uncertainties).

AR: That is correct. The statement will be revised.

AC: "The algorithm estimates the retrieval uncertainty, which quantifies the range of values that are feasible considering the uncertainty in the satellite measurements, auxiliary data and ORAC forward model."

RC: p 10, line 25: This statement differs from the statement at the end of Section 3.2 (phase is determined first to reduce computation time resulting from retrieving assuming both phases).

AR: That was the original processing setup, but in the end we decided to process both phases for all pixels. That was required in order to swap retrieval output if phase needed to be switched due to mismatches with CTT. Will clarify.

AC: p 8, line 25-27: "The main processor evaluates these inputs twice, assuming different cloud phases (e.g. ice and liquid). In theory, ORAC could use the preprocessed cloud mask and phase to select an appropriate method to reduce processing time."

RC: p 11, Section 4.1, Figure 3-5. The observation date/times should be stated here. I see they are listed in Section 4.3, but it is better to include them at first reference. Also, a thermodynamic phase image would be useful.

AR: Will add observation date/times here. Will also add the thermodynamic phase image.

AC: "The sample scene (07/22/2008 20:58 LST) is characterized by various cloud types, and the CC4CL cloud mask defines a relatively small fraction as cloud free (Figures 3 to 6). "

[Figure]

Figure 6. Cloud phase retrieval values for study area NA2 with data from AVHRR (left), MODIS (middle), and AATSR (right).

RC: p 11, lines 13-14: I'm guessing the peaks at 12 and 35 µm likely correspond to liquid and ice phase clouds, respectively.

AR: Agreed.

AC: "CER data are somewhat bimodal, having a primary peak at ~12 µm and a secondary peak at ~35 µm (Figure 07 and Table 06). These peaks probably correspond to liquid and ice phase clouds, respectively."

RC: p 11, line 17: The statement on cloud displacement here contradicts the statement in line 11.

AR: Although observation time difference is small, and thus cloud displacement, it cannot be discarded to contribute to the significance test, in particular to outliers.

AC: "Significance tests of mean differences and standard deviations of residuals between sensor retrievals are sensitive to outliers. Although cloud displacement due to observation time differences is probably small, we cannot discard its influence on such outliers."

RC: p 11, Section 4.1: What about relative radiometric calibration between the different sensors? Even minor differences of a couple percent could case large retrieval differences, particularly for COT.

AR: Agreed, will add a statement at the end of the paragraph.

AC: p 11, line 20: "Moreover, even modest relative radiometric calibration differences between sensors of a couple percent could cause large retrieval differences, particularly for COT."

RC: p 11, line 22: If median absolute CER uncertainty is 2µm, how does this correspond to a median relative uncertainty of 2% (line 24). Figure 10: What wavelengths are used for this RGB?

AR: The reviewer is correct, these statistics are wrong. Will correct. For the RGB, we used red = Ch4 solar component, green = Ch2, blue = Ch1.

AC: "Median absolute uncertainties are CTP = 26.7 hPa, COT = 6.1, CER = 2.0 µm, and cloud mask = 13.7 % (Figure 06). The median relative retrieval uncertainty (not shown) is relatively low for CTP and CER, but considerably larger for COT (CTP = 4.7 %, COT = 55.0 %, CER = 13.6 %). COT uncertainties increase with COT magnitude, and the RGB image (Figure 010, red = Ch4 solar

component, green = Ch2, blue = Ch1) shows that the largest uncertainties are found in cases of opaque cloud coverage and cloud over sea-ice surfaces."

RC: p 12, Section 4.3: Hard to call this "validation" without using a much larger dataset (e.g., months, seasons, years) for statistical analyses.

AR: Agreed, will rephrase.

AC: "Comparison with CALIOP"

RC: p 12, line 21: What assumptions are made other than adiabaticity (e.g., extinction profile, etc.)? Also, what does adiabaticity mean for an ice phase cloud?

AR: We assume that the cloud is vertically homogeneous with a constant lapse rate.

RC: p 12, Case Study NA1: Need to include the Figure number in the text.

AR: Agreed.

AC: "Study area NA1 is a completely cloud-covered scene over northern Canada containing clear and ice-covered land and open ocean surfaces (Figures 08 and 09)."

RC: p 13, lines 25-26: Which existing algorithms were compared to these results?

AR: Will remove the subordinate clause.

AC: "In general, the quantitative and qualitative agreement between CC4CL and CALIOP CTH is impressive."

RC: p 14, lines 10-11: Why not show the extensive validation here?

AR: Will add a reference to the Stengel paper mentioned above.

AC: "The results shown here are a representative sample from an extensive validation performed within the Cloud_cci project (Stengel et al., 2017)."

RC: p 15, lines 6-11: For the optimal estimation retrieval, are the spectral response differences handled similar to the ANN cloud mask (i.e., adjustment factors), or are they explicitly included in the forward model? What about relative radiometric calibration, could that be playing a role in the large MODIS-AATSR retrieval differences?

AR: Spectral response difference are taken into account when producing LUTs applied within CC4CL and are thus included in the forward model.

RC: p 15, line 18: Here calibration deficiencies are acknowledged. Relative calibration should be explored as a cause of the retrieval differences.

AR: Although we acknowledge that there are calibration differences, and doubt that sensors give precisely the same results, they were found to be consistent over vicarious calibration sites. For example, a 3 % offset between AATSR and MODIS has been found for visible channels (Smith and Cox, 2013), and a bias of < 0.3 K between MODIS and AVHRR longwave infrared channels (Cao and Heidinger, 2002). We think that this difference is not large enough to account for all the retrieval differences we see here. Note that the LUTs do take spectral differences into account, with the limitation that they have been calculated for an average value and not the full spectral shape, so that non-linear effects remain.

AC: "We did not quantify the contribution of each of these processes to overall retrieval differences when using different sensor data. In particular it would be worth investigating the impact of spectral response differences, which was outside the scope of this paper and the ESA Cloud_cci project."

D. L. Smith and C. V. Cox, "(A)ATSR Solar Channel On-Orbit Radiometric Calibration," in IEEE Transactions on Geoscience and Remote Sensing, vol. 51, no. 3, pp. 1370-1382, March 2013. doi: 10.1109/TGRS.2012.2230333

Changyong Cao, Andrew K. Heidinger, "Inter-comparison of the longwave infrared channels of MODIS and AVHRR/NOAA-16 using simultaneous nadir observations at orbit intersections", Proc. SPIE 4814, Earth Observing Systems VII, (24 September 2002); doi: 10.1117/12.451690; https://doi.org/10.1117/12.451690

RC: p 15, lines 29-30: Can the authors provide references for these user applications?

AR: Will add references.

AC: "On the one hand, they are useful for several user applications, such as model validation, data assimilation applications, or climate studies in general (Liu et al., 2017, Yang et al., 2016)."

Liu, C., R. P. Allan, M. Mayer, P. Hyder, N. G. Loeb, C. D. Roberts, M. Valdivieso, J. M. Edwards, and P.-L. Vidale (2017), Evaluation of satellite and reanalysis-based global net surface energy flux and uncertainty estimates, J. Geophys. Res. Atmos., 122, 6250–6272, doi:10.1002/2017JD026616.

Yang, Qinghua, et al. "Brief communication: The challenge and benefit of using sea ice concentration satellite data products with uncertainty estimates in summer sea ice data assimilation." The Cryosphere, vol. 10, no. 2, 2016, p. 761.

RC: p 16, line 29: "radiatively effective rather than physical cloud top"

AR: Will correct.

AC: "Any CTH retrieved from AVHRR (heritage) data is the radiatively effective rather than physical cloud top …"

RC: p 17, line 9: The MODIS C6 phase referred to here is that of the cloud optical properties algorithm, not the IR phase referred to earlier in the paper.

AR: The reviewer is correct. Our modifications above already account for that.

RC: p 18, lines 10-12: Perhaps this is worded poorly? I would imagine that real, complex vertical cloud structure is in fact a large source of retrieval errors, but the analytical approach to retrieval uncertainty used here (and in other retrievals) cannot account for this

AR: We think that, in the case of optically thick, i.e. opaque, clouds, the vertical cloud structure is not a major driver of TOA radiances and thus retrieval uncertainty. TOA radiances are mainly constrained by the cloud top layer, and also by lower layers until their influence becomes negligible due to vertical extinction.

AC: "Retrieval uncertainty is estimated using only well-understood error sources (e.g. measurement and forward model error), neglecting errors due to model assumptions (e.g. the complex, real vertical structure). Such errors can be approximated through validation activities and are not currently believed to be significant in most circumstances."

---

## Author Response (AR2)

**Reviewer #1**

The general impression from the response is that the authors are pretty much set on their approach. The use of ANNCOD in the background without quantifying its quality relative to CALIPSO is unfortunate, and it casts some doubt on the manuscript. I do think that a comparison of ANNCOD with CALIPSO makes sense, at least for COD<3. If this is what is used for the cloud mask, then it should be quality assessed. For bright regions, the authors' statement that passive sensors will always detect clouds for COD>1 (in their response) is incorrect. At some point, the authors comment that ANNCOD is a pseudo optical depth, which is "relative". But if that is the case, a formal definition should be provided what "pseudo" means. Following the current manuscript, ANNCOD seems to be unscaled, and trained with CALIPSO data, so it should reproduce CALIPSO data. However, the authors state that "ANNCOD does not provide any information on the absolute optical thickness value." At that point one has to wonder what it does do. If ANNCOD does not contain any information on COD, then the name "ANNCOD" is misleading, and the product should not be used as the basis for the cloud mask.

AR: We assume that CALIPSO correctly detects all clouds with COD > 1. Admittedly, this is not true for all passive sensors, in particular when only using the AVHRR heritage channel set. With that assumption in mind, we set all COD > 1 = 1 and thus do not account for the full spectrum of CALIPSO data. Moreover, the CALIPSO signal will be attenuated if thick clouds are present, and the lidar won't be able to penetrate to the surface if total column optical depth > 5. Thus, we think it is correct to refer to ANNCOD as a *pseudo* optical thickness. Although we trained the neural net with *true* CALIPSO COD data, our approach is no replacement of a Nakajima-King-style COD retrieval. ANNCOD is an unphysical quantity created by the neural net that is (a) observable and (b) correlated with cloud cover. An actual physical analogue is neither desired nor expected.

Otherwise, the added detail with regard to NN is helpful. However, questions remain how well it works with one single hidden layer. Just because the used IDL library does not provide more than one layer is not a good justification for limiting oneself to something that may not be adequate for the problem. Might this shallow NN be the reason that the authors don't show the validation against CALIPSO COD? In a similar vein, it is unfortunate that the authors do not want to show the histograms from figure 7 for snow bright surfaces (their response to the reviewer's comment to figure 7).

AR: We think that a single hidden layer is sufficient, and including one or more hidden layers would not produce significantly different results. See also:

K.M. Hornik, M. Stinchcombe, and H.White , „Multilayer feedforward networks are universal approximators", Neural Networks, vol. 4, no. 5, 1989, pp. 359-366,

which shows that a neural network with

- one hidden layer
- a sufficient number of nodes
- nonlinear activation functions

is capable of approximating any real-valued continuous scalar function.

We did comparisons between ANNCOD and CALIPSO COD, and analysed them as a function of illumination condition and surface type, including bright snow surfaces. Snow/ice covered surfaces indeed show a lower degree of agreement between the two entities, in particular regarding correlation coefficient and standard deviation ratio (Figure 1). In these cases, correlations range between 0.6 and 0.7, but are clearly larger for snow/ice-free surfaces during day and night (~0.85).

Although our methodology could still be improved (such as almost any other approach), we are confident that it provides reliable results of very good quality. Problems remain for bright land surfaces and twilight illumination conditions, which however have been addressed in a new NN cloud mask version 3.0 (Figure 2). Please note that the paper shows results for version 2.0 only. We are not hiding any negative results and are happy to share any information with the reviewer. Please see also the Product Validation and Intercomparison Report (Stengel et al., 2018) for a more detailed validation of the ANNCOD-derived cloud mask against CALIOP.

[Figure]

*Figure 1: Taylor plot of ANNCOD (version 2.0) versus CALIPSO COD for various illumination conditions and land cover types. The agreement between ANNCOD and CALIPSO COD is perfect if x = 1.0.*

[Figure]

*Figure 2: As above, but for version 3.0.*

A few minor comments:

p2,l49: "insufficient knowledge...propagates uncertainties..." still doesn't sound quite right. It is not the insufficient knowledge that propagates uncertainties. Just a language issue.

AR: Agreed, will change text.

AC: "and insufficient quantification of their state propagates uncertainties…"

p2,l78: "Consistency in approach can be traded for continuity of results, and multi-platform algorithms could exploit additional data when newer sensors become available" is still unclear and needs elaboration.

AR: Agreed, we will elaborate that sentence.

AC: "There is a trade-off between using the information from a single sensor optimally and using the information from different sensors consistently. The former may provide the most scientifically accurate data but often results in sharp discontinuities as instruments are introduced. Towards the aim of producing a stable, self-consistent climate record, this paper focuses on evaluating data from a range of sources through a unified methodology."

"For a single layer, optically thick (COT>1)" -- This is incorrect. Cloud with COD>1 are not optically

thick.

AR: We will rephrase that sentence.

AC: "For a single layer cloud with COT > 1,…"

Regarding the t-test (p13,l39): It is understood that readers will be familiar with a t-test, that was not the point of the comment.

AR: Our answer explains the purpose of the two-sample t-test, i.e. the null-hypothesis, and the meaning of the symbology. We are assessing whether the difference of the mean values of two independent sets of samples (drawn pairwise from AVHRR, MODIS, and AATSR data) is statistically significant. While we are aware of the statistical limitations of the t-test and its frequent misuse, this is a clear, simple, and frequently applied test to punctuate a simple point. As we do not aim to infer causation from these differences, and would prefer not to lecture the reader in an already long paper, we leave our text unchanged.

p13,l45: "spatiotemporally collocated sensors" The comment was not about the mathematical meaning of the intercomparison, but about the physical meaning. Why *are* the data different, even though they are collocated?"

AR: There are various reasons why the data are different despite collocation, such as sub-pixel cloudiness (the cloud varies on a scale smaller than that observed, so slight differences in pixel position produce statistically significant changes in value), cloud displacement between observation times, and differences in spectral response functions. Repeating the analysis on an even coarser grid would not improve any of these issues.

**Reviewer #2**

p 2, lines 34-35: I think the authors inserted the Marchant reference and description in the wrong sentence. That paper discusses the MODIS C6 optical property phase, not cloud detection/masking, and fits into the next sentence. Moreover, the next sentence discussing phase should include the Baum et al (2012) reference also since it is referring to the MODIS C6 IR phase which uses 8.5 and 11µm in addition to 7.3 and 12µm (the authors provide an incorrect description of this). There are two independent phase algorithms in MODIS C6 and both should be referenced.

AR: Agreed, we will add references as suggested.

AC: "For cloud masking, the retrieval frameworks apply various approaches such as Naïve Bayes (PATMOS-X), dynamic thresholding (CLARA-A1), or a battery of threshold test (MODIS C6). Finally, cloud phase or type is determined as a function of a combined convergence/cloud top temperature (CTT)-test (CLARA-A1), the Pavolonis et al. (2005) threshold algorithm (PATMOS-X), or a majority vote algorithm that combines four phase tests based on CTT, tri-spectral IR, 1.38 µm, and spectral CER data (Marchant et al., 2016, Baum et al., 2012)."

p 3, lines 6-7: "and beyond a penetration depth into the cloud corresponding to > 1 cumulative optical depth (Baum et al., 2012)." I'm not sure how this in itself describes an insensitivity to cloud top properties. The roughly 1 optical depth penetration depth is simply the reason why the MODIS C6 cloud top retrieval (and any other IR cloud top retrieval) loses sensitivity for optically thin clouds and why it cannot be considered the physical cloud top.

AR: Agreed, we will remove that sentence.

AC: "Still, the MODIS C6 cloud top retrieval loses sensitivity for optically thinner clouds (COT < 2, Menzel et al. (2010); Christensen et al. (2013))."

p 3, line 22: "However, the detection of optically thin ice clouds over warm, bright surfaces remains problematic (Marchant et al., 2016)." This should actually refer to the phase identification of optically thin ice clouds, which is what the Marchant paper discusses, not the detection, which is a cloud mask issue.

AR: Agreed, we will rephrase in order to refer to the correct product.

AC: "…, and the phase identification has been improved for liquid clouds. However, the phase identification of optically thin ice clouds over warm, bright surfaces remains problematic (Marchant et al., 2016)."

p 5, line 21: CALIOP is the lidar itself, CALIPSO is the satellite platform.

AR: Agreed, we will correct.

AC: "The Aqua satellite is a member of the "A-Train" constellation, which also includes the CALIPSO and CloudSat satellites."

p 6, line 8: "on-board" instead of "on-bard"

AR: Will correct, also removing repeated "is" in that sentence.

AC: "ATSR is equipped with on-board calibration capabilities, …"

p 16, lines 15-16: "In particular it would be worth investigating the impact of spectral response differences, which was outside the scope of this paper and the ESA Cloud_cci project." Adding this statement is fine, but it is now at odds with previous statements (e.g., same page lines 1-2) that spectral response differences are unlikely the causes of retrieval differences, and with the immediately following statement.

AR: Agreed, we will tone down that sentence to be consistent with our previous and following statements that spectral response differences are not a main driver.

AC: "Although we found that spectral response differences are probably negligible, we still find it worth quantifying their impact, which was outside the…"